# Structural and molecular determinants of *Candida glabrata* metacaspase maturation and activation by calcium

Léa Conchou[1], Bastien Doumèche[2], Frédéric Galisson[1], Sébastien Violot [1], Chloé Dugelay[1], Eric Diesis[3], Adeline Page [3], Anne-Lise Bienvenu[2,4], Stéphane Picot [2,5], Nushin Aghajari [1✉] & Lionel Ballut [1✉]

Metacaspases are caspase-like homologs which undergo a complex maturation process involving multiple intra-chain cleavages resulting in a composite enzyme made of a p10 and a p20 domain. Their proteolytic activity involving a cysteine-histidine catalytic dyad, show peptide bond cleavage specificity in the C-terminal to lysine and arginine, with both maturation- and catalytic processes being calcium-dependent. Here, we present the structure of a metacaspase from the yeast *Candida glabrata, Cg*MCA-I, in complex with a unique calcium along with a structure in which three magnesium ions are bound. We show that the $Ca^{2+}$ ion interacts with a loop in the vicinity of the catalytic site. The reorganization of this cation binding loop, by bringing together the two catalytic residues, could be one of the main structural determinants triggering metacaspase activation. Enzymatic exploration of *Cg*MCA-I confirmed that the maturation process implies a *trans* mechanism with sequential cleavages.

[1] Molecular Microbiology and Structural Biochemistry, UMR 5086, CNRS-Université de Lyon, F-69367 Lyon, France. [2] Université de Lyon, Université Lyon 1, Institut de Chimie et Biochimie Moléculaires et Supramoléculaire, ICBMS UMR 5246, CNRS, F-69622 Lyon, France. [3] University of Lyon, INSERM, ENS Lyon, CNRS, Protein Science Facility, SFR BioSciences, UAR3444/US8, F-69366 Lyon, France. [4] Service Pharmacie, Groupement Hospitalier Nord, Hospices Civils de Lyon, F-69004 Lyon, France. [5] Institute of Parasitology and Medical Mycology, Hôpital de la Croix-Rousse, Hospices Civils de Lyon, F-69004 Lyon, France. ✉email: nushin.aghajari@ibcp.fr; lionel.ballut@ibcp.fr

Regulated cell death (RCD) is a physiological phenomenon essential for survival and maintenance of cell homeostasis in multicellular organisms. This is a highly regulated process in animals, that depends on a cellular machinery and notably caspases (cysteine-containing aspartate-specific proteases)[1]. The identification of RCD in non-metazoan organisms[2] was followed by the discovery of caspase-like structural homologs termed metacaspases. Unlike caspases, which are found in animals and in some viruses, metacaspases are present in plants, protozoa, and yeast, but are absent in animals[3]. They are involved in yeast programmed cell death in response to oxidative stress[4], ageing[5] or viral toxins[6] and also non-death functions as cell cycle regulation[7] and in protein quality control (PQC) for clearance of protein aggregates[8].

Caspases and metacaspases are both classified into the C14 protein family within the CD clan of cysteine proteases[9], are synthesized as inactive zymogens, and share a structural hemoglobinase fold composed of the p10 and p20 domains, containing the catalytic His/Cys dyad. While metazoan executioner caspases are activated through their processing by initiator caspases and their dimerization, metacaspases do not undergo dimerization and are active as monomers[3]. They are moreover specific to Arg and Lys residues in the P1 position[10]. Metacaspases are further sub-classified in three types according to the architecture of the p10 and p20 domains with type I containing an additional N-terminal domain[10]. Also, the presence of calcium is required for their activity and further autocatalytic processing[10–12]. As in animals, yeast cell-death has been linked to an increase of the intra-cellular and intra-mitochondrial $Ca^{2+}$ concentration, where the latter plays a role in death signal transduction and typical morphological changes[13]. Previously, the three-dimensional structure of *Trypanosoma brucei* metacaspase type I has been solved in complex with $Sm^{3+}$ suggesting the presence of a $Ca^{2+}$ binding site in the p20 domain inducing structural changes of a loop near the active site[14]. In addition, the study of *Arabidopsis thaliana* type II metacaspases, resulted in the identification of a $Ca^{2+}$-dependent multi-cleavage process where metacaspases are thought to transduce $Ca^{2+}$ signals and activate cell death pathways[15]. However, the enzymatic and structural basis for $Ca^{2+}$ activation of metacaspases remains to be elucidated.

In this work, we report the determination of the second crystal structure of a yeast metacaspase namely that from *Candida glabrata*, *Cg*MCA-I. We observed electron density corresponding to a calcium ion, thereby allowing the identification of a calcium-binding site, in the vicinity of a previously observed samarium ion binding site[14]. Through a functional, enzymatic and structural study, the $Ca^{2+}$ dependent mechanism of maturation and activation of type I metacaspases has been investigated. Throughout, the term maturation is used to describe the different proteolytic cleavages that the metacaspase will undergo and, by extension, the different intermediates of maturation termed $E_0$ to $E_3$. The term activation corresponds to the increased catalytic activity of the metacaspase which harbors either low activity during the new step of maturation "$E_{low}$", or high activity at the last step of maturation "$E_{high}$".

## Results

**Cation-dependent maturation of *Cg*MCA-I.** To decipher the molecular determinants of the maturation process of *Cg*MCA-I, the full-length (residues 1–392) wild-type enzyme was over-expressed in *E. coli* and purified by affinity chromatography and gel filtration. Size exclusion chromatography indicates that, the elution volume of *Cg*MCA-I corresponds to an apparent mass of 38 kDa with an additional peak at 10 kDa suggesting that the protein could be submitted to a maturation process as observed

for other metacaspases[14–16] (Supplementary Figs. 1, 2). It has been shown that calcium may facilitate and, in some cases, is mandatory[12,14,16,17] for maturation of the enzyme. In order to gain a better understanding of *Cg*MCA-I maturation, we purified by affinity chromatography using Biosprint 96 (see Maturation Assay, SI) and analysed by gel electrophoresis the wild-type protein after 0, 4 and 8 days, respectively, in the presence or the absence of calcium or EGTA to observe the maturation process over time and the impact of $Ca^{2+}$ ions on this process (Fig. 1).

In the absence of calcium or EGTA, *Cg*MCA-I undergoes cleavage and starts to maturate at day 0, continuing until day 8 without reaching a complete mature form namely the $E_3$ form (*vide infra*). Different intermediary bands appear over time with sizes from 10 kDa to 38 kDa. The maturation in the presence of $Ca^{2+}$ is clearly accelerated and at day 8, a thin band at 23 kDa is visible which could be attributed to the p20 domain. In the presence of EGTA, the process is stopped at day 0, confirming the need for a divalent cation in the maturation process. In addition to the full-length metacaspase at 48 kDa, a thinner band is observed at 38 kDa indicating that maturation could start during expression and purification prior to EGTA addition (Fig. 1b).

The fact that maturation is visible in the absence of additional $Ca^{2+}$ after purification could indicate that calcium partially binds to the protein during expression or that another divalent cation could be present, even at low concentration, and could be bound prior to purification. To confirm the role of calcium and/or other divalent cations as activators, we tested three additional divalent metal ions: $Mn^{2+}$, $Mg^{2+}$, and $Zn^{2+}$ (Fig. 1c). In the presence of $Mn^{2+}$, the enzyme shows a behavior similar to that observed for maturation in the absence of divalent cations. More surprisingly, in the presence of $Mg^{2+}$ or $Zn^{2+}$, the maturation process seems to be blocked at the very first cleavage event such as in the presence of EGTA suggesting that these two divalent cations could play an inhibiting role on the maturation (Fig. 1c).

**Analysis of amino-acid residues involved in *Cg*MCA-I maturation.** Analysis of metacaspase fragments by nano-reversed phase liquid chromatography coupled to high-resolution mass spectrometry (nanoLC-HRMS) with an Orbitrap analyzer was used to determine the cleavage sites with higher accuracy. The high resolution of the orbitrap mass spectrometer allows determining, after deconvolution of the multicharged MS spectrum, the molecular weight of the different cleaved fragments of *Cg*MCA-I with high accuracy (Supplementary Fig. 3). We identified residues $Arg^{307}$, $Lys^{263}$, and $Arg^{54}$, as the first, second and third cleavage sites, respectively (Fig. 1b).

In order to decipher the molecular determinants of the maturation process, a series of cloning and maturation experiments were performed, and subsequently the time-course maturation without additives or in the presence of calcium or EGTA was explored. The first aim was to determine the initial cleavage site. For this purpose, *Cg*MCA-I was cloned in a pET52 vector to introduce a His-Tag at the C-terminal (*Cg*MCA-I$^{His}$) and then observed following 0, 4, and 8 maturation days without additives by gel electrophoresis and (Fig. 2a) by western blotting (Fig. 2b). A band around 10 kDa appears at day 0 for *Cg*MCA-I$^{His}$ but only at day 4 for $^{His}$*Cg*MCA-I, indicating that the first cleavage occurs at the C-terminal part of the protein. In addition, the lack of band at 49.7 kDa at day 4 for *Cg*MCA-I$^{His}$ indicates that the release of the C-terminal occurs in the very first steps of maturation. On the other hand, the presence of a band at 48 kDa in the western blot until day 8 suggests that the cleavage of the N-terminal part of the protein occurs at the very end of the process (Fig. 2a–c). To confirm this observation, the R54A mutant was expressed and its maturation analyzed

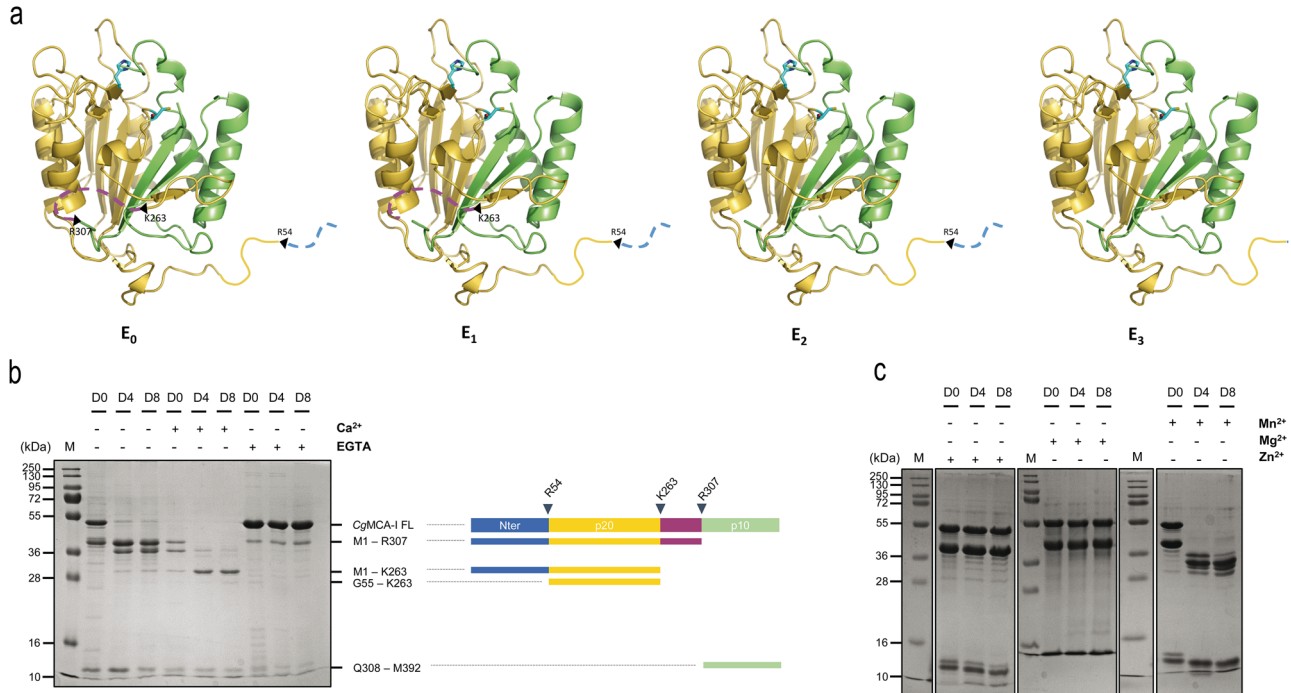

**Fig. 1 Maturation of *Cg*MCA-I in the presence or absence of divalent cations and EGTA. a** Structural representation of maturation intermediate forms of *Cg*MCA-I starting from the non-matured form ($E_0$), after the first and second cleavage ($E_1$ and $E_2$ respectively), and the mature form ($E_3$). **b** SDS-PAGE of *Cg*MCA-I aliquots at maturation day 0 (D0), 4 (D4) or 8 (D8) in the presence or absence of $Ca^{2+}$ (10 mM) or EGTA (1 mM). The band at 48 kDa corresponds to *Cg*MCA-I full length (*Cg*MCA-I FL). Schematic representation of the maturation process of *Cg*MCA-I with the N-terminal part, the p20 domain, the linker, and the p10 domain, respectively colored in *blue*, *yellow*, *purple*, and *green*. The proteolytic cleavage after residues R307, K263 and R54 results in the progressive appearance of maturation intermediate forms which correspond to different bands identified by their residues on the right as determined by mass spectrometry. **c** SDS-PAGE of *Cg*MCA-I aliquots at D0, D4, and D8 in the presence or the absence of $Mn^{2+}$ (10 mM), $Mg^{2+}$ (10 mM), and $Zn^{2+}$ (10 mM).

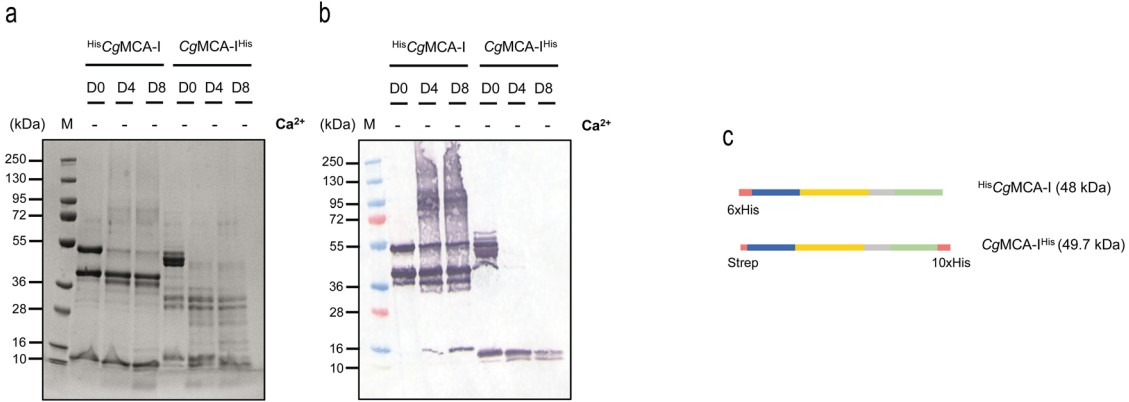

**Fig. 2 Maturation of $^{His}$*Cg*MCA-I and *Cg*MCA-I$^{His}$.** SDS-PAGE (**a**) and Western-blot (**b**) analysis of aliquots of $^{His}$*Cg*MCA-I and *Cg*MCA-I$^{His}$ constructs at D0, D4, and D8 in the absence of $Ca^{2+}$. (**c**) Schematic representation of $^{His}$*Cg*MCA-I and *Cg*MCA-I$^{His}$ constructs.

(Supplementary Fig. 4a). It appears that in the presence of calcium the process is delayed, leading only to intermediate forms of maturation without reaching a fully mature form of *Cg*MCA-I after eight days, and confirming that cleavage at position R54 is most likely the last one to occur.

In order to characterize the second and first cleavage sites which should be at positions 263 and 307, respectively, Lys$^{263}$ and Arg$^{307}$ were mutated to an alanine (K263A and R307A) and the maturation process followed (Supplementary Fig. 4b, c). Surprisingly, both mutants seem to undergo cleavages with a maturation process mimicking the wild-type enzyme, hence Lys$^{263}$ was mutated to an aspartate and to a phenylalanine, respectively, to induce more drastic effects (Supplementary Fig. 4d, e). In both

cases, and in the lack of calcium, the first cleavage event was observed at day 0 with the appearance of a band at 38 kDa. However, in the course of maturation in the days following, an aberrant process seems to occur with a number of secondary cleavage sites resulting in the disappearance of the protein from day 4. In the presence of calcium, only a thin band with a size which could correspond to the mature form was observed. This suggests that the mutants most probably underwent aberrant maturation, possibly partially causing their degradation (Supplementary Fig. 4d, e). Altogether, this points to the presence of basic residues not being mandatory at these positions, provided that the residue replacing the lysine does not drastically change the function held by the side chain.

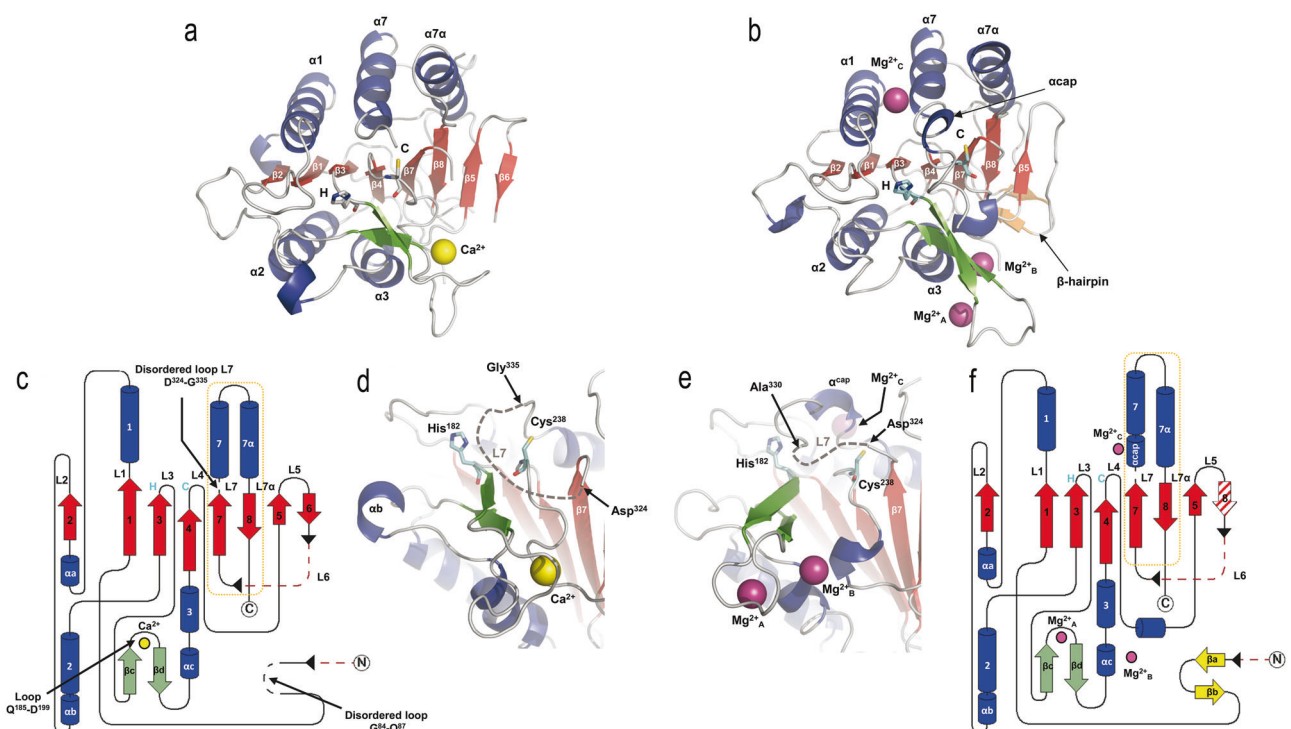

**Fig. 3 Overall structure organization of *Cg*MCA-I.** Three-dimensional structures of *Cg*MCA-I$^{Ca}$ and *Cg*MCA-I$^{Mg}$ monomers, respectively, in complex with (**a**) Ca$^{2+}$ (*yellow sphere*) or (**b**) Mg$^{2+}$ (*purple spheres*). α-Helices (*blue*) and β-strands (*red*), with the numbering of the secondary structure elements shown in *white* and *black*, respectively (**a**). The three Mg$^{2+}$ ions present in *Cg*MCA-I$^{Mg}$ are named Mg$^{2+}_A$, Mg$^{2+}_B$, and Mg$^{2+}_C$ (**b**). Catalytic residues are highlighted as stick presentations by H (His$^{182}$) and C (Cys$^{238}$) and colored in *cyan*. Topological representation of *Cg*MCA-I$^{Ca}$ (**c**) and *Cg*MCA-I$^{Mg}$ (**f**). Disordered parts of the proteins are shown in *black* dotted lines and missing parts of the protein due to the maturation process are shown in *red* dotted lines (**c**, **f**). Cleavage sites are indicated by *black* arrow heads and the p10 subunits are framed by an *orange* dotted line (**c**, **f**). Loops numbering (L1 to L7/L7α) are indicated (**c**, **f**). Catalytic site organization of *Cg*MCA-I$^{Ca}$ (**d**) and *Cg*MCA-I$^{Mg}$ (**e**), showing the position of the two catalytic residues (His$^{182}$ and Cys$^{238}$) respective to the cations binding sites and the disordered L7 loop, for which the disordered part is indicated by *gray* dotted lines.

**Overall structural features of *Cg*MCA-I**. To elucidate the molecular determinants for cation-dependent metacaspase activation, *Cg*MCA-I was expressed in *E. coli* and purified to homogeneity without additives or in the presence of 10 mM Ca$^{2+}$ or of 1 mM EGTA in order to obtain non-maturated, partially or fully maturated enzyme for crystallization. Alternatively, we used a purified C238A catalytic mutant in order to obtain a non-maturated protein.

Crystallization assays were carried using enzymes at day 0 and day 4 following purification. Depending on the time of maturation, crystals grew either from a solution containing 0.2 M (NH$_4$)$_2$SO$_4$, 30% (*w/v*) PEG 4000 (day 0 protein) or a solution containing 0.2 M magnesium acetate, 20% (*w/v*) PEG 3350 (day 4 protein).

Structures from these two crystal forms were determined by the molecular replacement method using the structure of the homologue from *Saccharomyces cerevisiae*, *Sc*MCA-I (PDB code: 4F6O[16]), as search model. The crystal structures displayed clear electron density for Ca$^{2+}$ and Mg$^{2+}$ ions, and were named according to this issue, *Cg*MCA-I$^{Ca}$ (PDB code: 7QP1) or *Cg*MCA-I$^{Mg}$ (PDB code: 7QP0), respectively (Fig. 3).

*Cg*MCA-I$^{Ca}$ crystallized in the trigonal space group *P*3$_2$ with two molecules in the asymmetric unit (Table 1, Supplementary Fig. 5a), whereas *Cg*MCA-I$^{Mg}$ crystallized in the monoclinic space group *C*2 with two molecules in the asymmetric unit (Table 1, Supplementary Fig. 5b). The structures were solved to respectively 3.0 Å and 1.6 Å resolution and share common features. The monomers *Cg*MCA-I$^{Ca}$ and *Cg*MCA-I$^{Mg}$ both contain a p20 and a p10 domain forming a caspase-like core in which a two-stranded anti-parallel β-sheet from the p10 domain completes a

six stranded parallel β-sheet in the p20 domain (Fig. 3a–c, f). This central β-sheet is surrounded by three helices on one side (α1, α4, and α5) and two α-helices on the other side (α2 and α3) (Fig. 3a–c, f). In addition, a β-hairpin (βc and βd) between β3 and β4 complete the p20 domain (Fig. 3c, f). Cys$^{238}$ and His$^{182}$ are located on loops *Cg*L4 and *Cg*L3, respectively, from the catalytic dyad (Fig. 3a–c, f).

Regarding the specificities of *Cg*MCA-I$^{Ca}$, three short helices (αa, αb and αc) complete the p20 domain (Fig. 3a, c). In addition, as observed in the crystal structure of *Sc*MCA-I[16], loop *Cg*L7 (named L3 in *Sc*MCA-I), which may cap the substrate-binding groove, is disordered from Asp$^{324}$ to Gly$^{335}$ (Fig. 3c, d, Supplementary Fig. 6). As opposed to *Sc*MCA-I, a short loop formed by residues Gly$^{84}$ to Gln$^{87}$, displays no electron density (Fig. 3c). Surprisingly, the two monomers of the *Cg*MCA-I$^{Ca}$ dimer observed in the asymmetric unit interact *via* a β-sheet complementation between the β6 and β5' strands (Supplementary Fig. 5a). This unexpected organization being similar to that of the dimer usually observed for caspases, SAXS experiments were performed to determine the oligomeric state of *Cg*MCA-I in the presence of calcium and confirmed that this enzyme is monomeric in solution (Supplementary Fig. 7).

Regarding the maturation process, the N-terminal and the linker located between p20 and p10 domains and which corresponds to the *Cg*L6 loop, are both cleaved and residues Met$^1$ to Gly$^{68}$ and Asn$^{260}$ to Ile$^{309}$ show no electron density in the 2Fo-Fc map (Fig. 3c, Supplementary Fig. 8a, b). Due to the flexibility at the cleaved ends, we were not able to determine with accuracy the residues after which the cleavage occurred only based on the structure, however, the boundaries of the missing

**Table 1 Data collection and refinement statistics.**

| Structure-ID | CgMCA-I$^{Ca}$ | CgMCA-I$^{Mg}$ |
|---|---|---|
| PDB entry | 7QP1 | 7QP0 |
| **Data collection** | | |
| Beamline | PXIII | PROXIMA1 |
| Wavelength (Å) | 0.99999 | 0.97857 |
| Space group | P32 | C2 |
| **Cell dimensions** | | |
| a, b, c (Å) | 97.4 97.4 54.6 | 95.7 83.6 87.8 |
| α, β, γ (°) | 90.0 90.0 120.0 | 90.0 106.6 90.0 |
| Resolution range (Å) | 40.0–3.0 | 40.0–1.6 |
| Total reflections | 123226 | 303345 |
| Unique reflections | 11590 | 87056 |
| R$_{meas}$ (%) | 9.9 (101.1) | 10.3 (85.9) |
| CC$_{1/2}$ (%) | 99.9 (82.3) | 99.7 (76.0) |
| I/σ(I) | 20.5 (2.5) | 8.44 (1.84) |
| Multiplicity | 10.6 (10.7) | 3.5 (3.4) |
| Completeness (%) | 100 (100) | 99.6 (99.7) |
| No. mol. /asymm. unit | 2 | 2 |
| **Refinement** | | |
| R$_{work}$/R$_{free}$ (%) | 20.78/26.05 | 17.53/20.23 |
| **No. atoms** | | |
| Protein | 4014 | 4111 |
| Ligand/ion | 4 | 36 |
| Water | 7 | 627 |
| **Average B-factor (Å$^2$)** | | |
| Protein | 93.2 | 25.6 |
| Ligand/ion | 74.0 | 50.2 |
| Water | 74.2 | 39.6 |
| **r.m.s.d.** | | |
| Bond lengths (Å) | 0.006 | 0.017 |
| Angles (°) | 1.120 | 1.618 |
| **Ramachandran** | | |
| Favored (%) | 90.5 | 98.8 |
| Allowed (%) | 8.7 | 1.2 |
| Outliers (%) | 0.8 | 0.0 |

part are in perfect accordance with the cleavage site identified by mass spectrometry (Supplementary Fig. 3).

Regarding the specificities CgMCA-I$^{Mg}$, and as opposed to CgMCA-I$^{Ca}$, no β-strand complementation was present to stabilize the dimer which instead shows contacts between helix 1 and loop CgL2 (Supplementary Fig. 5b). The overall structure is highly similar to that of CgMCA-I$^{Ca}$ with a RMSD of 0.823 Å calculated on all Cα atoms. As for CgMCA-I$^{Ca}$, a fully maturated protein was observed, but with the N-terminal (Met$^1$ to Ser$^{72}$) and linker (Asn$^{260}$ to Ile$^{309}$—a part of CgL6) missing (Fig. 3f, Supplementary Fig. 8c, d). As opposed to CgMCA-I$^{Ca}$, residues Gly$^{84}$ to Gln$^{87}$ show clear electron density and form a hairpin stabilized by two short β-strands (βa and βb) (Fig. 3b, f, Supplementary Fig. 8e, f). A disordered CgL7 loop is still observed, but the segment from Asp$^{330}$ to Gly$^{335}$ could be constructed. Interestingly, this longer structured segment forms a one-turn helix (Gly$^{335}$-Asn$^{337}$, αcap) and caps the substrate-binding groove limiting, under this conformation, the access to the catalytic site (Fig. 3e).

**Structural basis for CgMCA-I Ca$^{2+}$ dependent maturation and activation.** Several studies have shown that the maturation process of metacaspases (AtMCA-IIb[17], AtMCA-IId[12], TbMCA-Ib[14], ScMCA-I[16]) is calcium-dependent. To solve the structure of TbMCA-Ib[14], the authors used samarium, a calcium-mimicking lanthanide. They reported that the presence of Ca$^{2+}$ always hindered crystal formation and that Ca$^{2+}$ could not be detected after the soaking experiment. More recently, a three-dimensional

structure of a plant metacaspase[15] was solved and a soaking experiment carried out in the presence of Ca$^{2+}$. Here, again and despite clear conformational reorganization of some parts of the protein, no electron density was observed for Ca$^{2+}$.

To determine the structure of CgMCA-I$^{Ca}$, we purified the wild-type metacaspase in the presence of 10 mM CaCl$_2$ and performed crystallization assays at day 0 using a non-fully maturated protein. Crystals were observed only after three months, and the crystal structure reveals that the maturation process occurred in the course of the crystallization. We observed a clear electron density for Ca$^{2+}$ (Figs. 3a, d, 4, Supplementary Fig. 9). Interestingly, the Ca$^{2+}$ ion is in the vicinity of the Sm$^{3+}$ binding site as observed in TbMCA-Ib. In CgMCA-I$^{Ca}$ (Fig. 5a), Ca$^{2+}$ is hepta-coordinated by a water molecule and residues Gly$^{196}$, Asp$^{198}$ (Asp$^{173}$ in TbMCA-Ib), Asp$^{214}$ (Asp$^{189}$ in TbMCA-Ib), Thr$^{242}$ and Asp$^{245}$ (Asp$^{220}$ in TbMCA-Ib) (Supplementary Fig. 5a, b). Similarly, Sm$^{3+}$ in TbMCA-Ib is hepta-coordinated but shows a different binding mode[14]. Indeed, except for the three conserved aspartates (Asp$^{173}$, Asp$^{189}$ and Asp$^{220}$), the rest of the coordination is ensured by two water molecules and Asp$^{190}$ (Fig. 5b). In addition, Arg$^{171}$ and Ser$^{217}$ in TbMCA-Ib which correspond to Gly$^{196}$ and Thr$^{242}$ in CgMCA-I$^{Ca}$, respectively, are not involved in the cation binding (Fig. 5b). Altogether, this original Ca$^{2+}$ coordination results in a complete reorganization of segment Gly$^{183}$-Ile$^{201}$ and its associated loop Gln$^{185}$-Asp$^{199}$ (Fig. 5a).

First, Ca$^{2+}$ is slightly displaced (3.4 Å) compared to Sm$^{3+}$ (Fig. 5a). Second, a TbMCA-Ib disordered loop (Cys$^{162}$-Asp$^{173}$) is stabilized in CgMCA-I$^{Ca}$ (loop Gln$^{185}$-Asp$^{199}$) due to the coordination by Gly$^{196}$ and Asp$^{198}$ (Supplementary Fig. 10a, b). This loop conformation shows clear electron density in which the calcium is capped. Importantly, this loop is part of a small domain made of two small antiparallel β-strands and one short helix, a domain which immediately succeeds the loop CgL3 bearing the catalytic histidine (Fig. 3c). Third, the segment Asp$^{236}$-Asp$^{245}$ (Asp$^{211}$-Asp$^{220}$ in TbMCA-Ib) which holds the catalytic cysteine is stabilized by Thr$^{242}$ and Asp$^{245}$ (Fig. 4, Supplementary Fig. 10b). Interestingly, the "cation binding loop" observed in ScMCA-I (Gln$^{223}$-Asp$^{237}$) and in TbMCA-Ib (Cys$^{162}$-Asp$^{173}$) is ordered in CgMCA-I$^{Ca}$ which suggests that the calcium has probably a role of stabilization not only for loop Asn$^{185}$-Asp$^{199}$ but also for segment Asp$^{236}$- Asp$^{245}$, the whole contributing to bringing together the two catalytic residues (Fig. 4).

To solve the structure of CgMCA-I$^{Mg}$, we performed experiments similar to those described above. Nevertheless, after purification, the enzyme maturated during 4 days prior to crystallization assays. For the crystals used for structure determination, the crystallization condition included 200 mM magnesium. The electron density map revealed the presence of three magnesium ions being hexa-coordinated (Fig. 5c, d, Supplementary Fig. 10c–e). One of the Mg$^{2+}$ ions is positioned at the very same place as the samarium ion observed in the crystal structure of TbMCA-Ib but instead of being coordinated by four amino-acid residues and a water molecule as observed for this latter, in CgMCA-I$^{Mg}$ it is coordinated by five water molecules and Asp$^{214}$ (Asp$^{189}$ in TbMCA-Ib) (Fig. 5d). Interestingly, most of the aspartates coordinating samarium in TbMCA-Ib are conserved in CgMCA-I$^{Mg}$ and interact with three of the water molecules coordinating magnesium, namely Asp$^{198}$ (Asp$^{173}$ in TbMCA-Ib), Asp$^{215}$ (Asp$^{190}$ in TbMCA-Ib) and Asp$^{245}$ (Asp$^{220}$ in TbMCA-Ib). In addition, Asp$^{199}$ and the main chains of Thr$^{242}$ and Met$^{197}$ interact with the two last water molecules (Fig. 5b, d). Altogether, it appears that Sm$^{3+}$ most probably mimics Mg$^{2+}$ rather than Ca$^{2+}$. The fact that samarium is observed in non-maturated TbMCA-Ib could explain the different orientations observed for the cation binding loops of the two enzymes (Fig. 5a, c). In CgMCA-I$^{Mg}$,

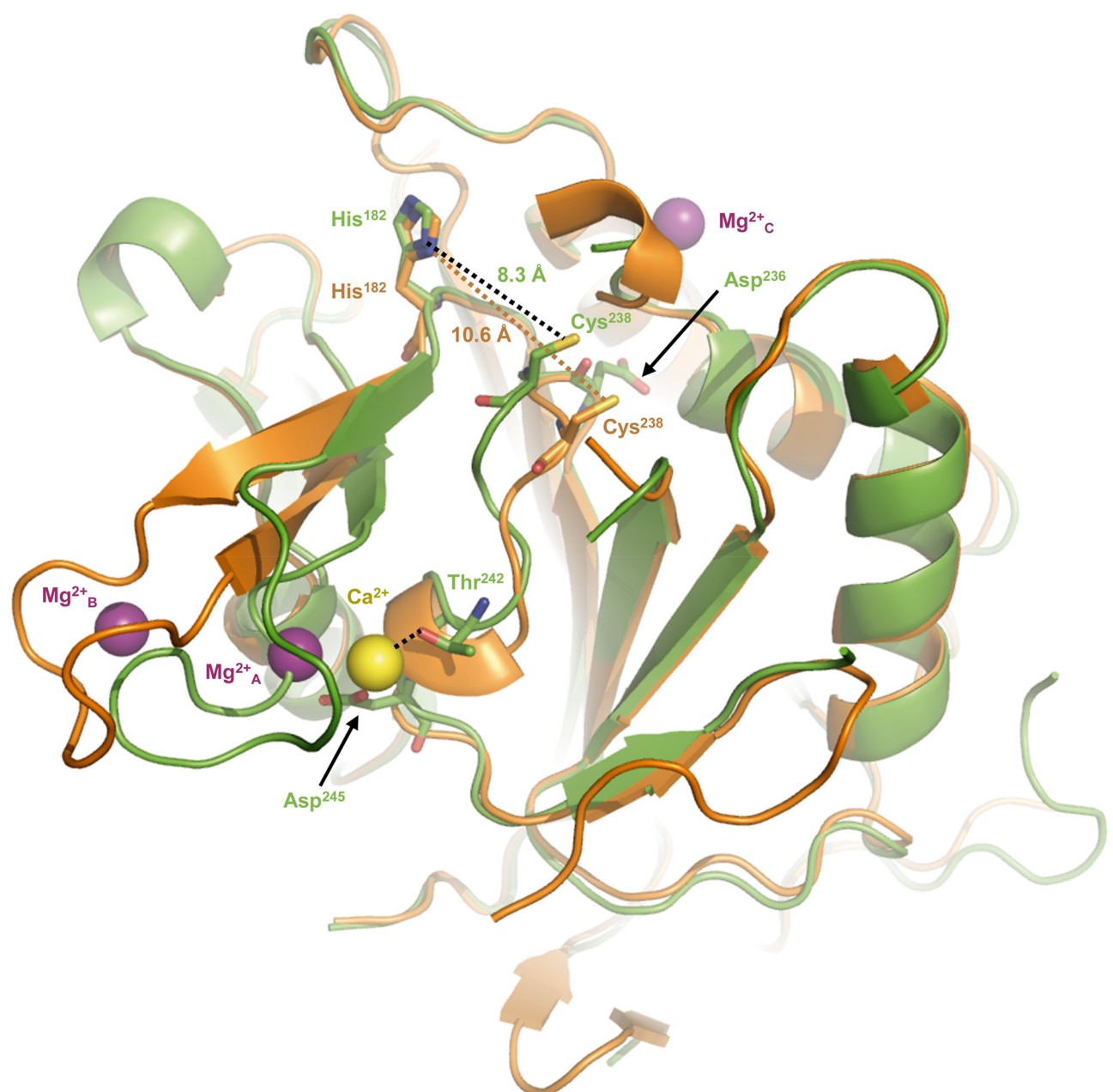

**Fig. 4 Reorganization of the catalytic sites of *Cg*MCA-I$^{Ca}$ and *Cg*MCA-I$^{Mg}$.** The distance between the thiol group of the catalytic cysteine Cys$^{238}$ and the imidazole nitrogen of the catalytic histidine His$^{182}$ in the two crystal structures, with that of *Cg*MCA-I$^{Ca}$ shown in *green* and *Cg*MCA-I$^{Mg}$ in *orange*. Ca$^{2+}$ and Mg$^{2+}$ ions are shown as *yellow* and *purple* spheres, respectively.

the enzyme is fully maturated but in parallel, we have shown that Mg$^{2+}$ seems to impair the maturation of the metacaspase. Interestingly, in the presence of Mg$^{2+}$ the distance between the two catalytic residues is 10.6 Å, against 8.3 Å when in presence of Ca$^{2+}$, suggesting that the presence of calcium forces the catalytic site to adopt a proper organization for its autolysis activity (Fig. 4). A second magnesium ion coordinated by two acidic residues (namely Asp$^{188}$ and Asp$^{190}$) and three water molecules is present in the "calcium capping loop" contributing to its stabilization and reorientation (Supplementary Fig. 10c, e).

**Enzymatic analysis of *Cg*MCA-I maturation.** In preliminary experiments, apparent kinetic constants of *Cg*MCA-I were determined in the presence of 10 mM calcium at days 0 and day 8

of maturation using Z-GGR-AMC as substrate (concentration range of 0–50 μM) (Supplementary Fig. 11). A kinetic model based on the enzymatic steady-state hydrolysis of Z-GGR-AMC including chemical hydrolysis was used to determine kinetic parameters (Supplementary Table 5).

The first-order kinetic constant for the non-catalyzed ($k_{chem}$) reaction is comprised between $0.1183+/-~0.0092~min^{-1}$ and $0.107+/-~0.012~min^{-1}$. As expected, the enzymatic reaction follows Michaelis-Menten behaviour in the substrate concentration range assayed. At day 0, the $k_{cat}$ value is $2.8+/-~0.7~min^{-1}$ and reaches $15.8+/-~0.6~min^{-1}$ at day 8 indicating that the matured metacaspase is 5 to 6 times more active than the non-matured form. Interestingly, the $K_m$ value for Z-GGR-AMC (18 μM) does not change during maturation, which could explain why the substrate binding site is only slightly affected. This

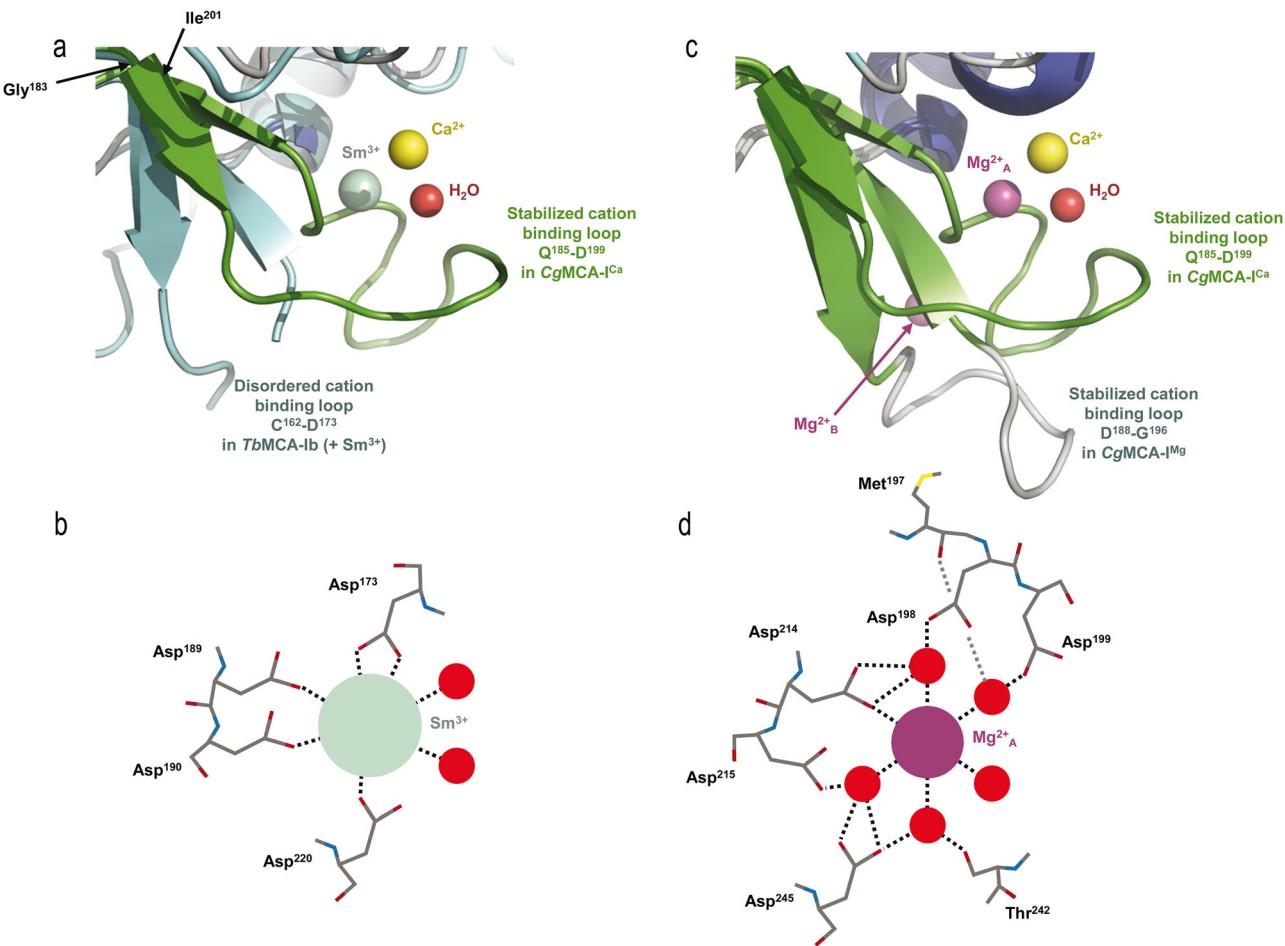

**Fig. 5 Structural organization of *Cg*MCA-I$^{Ca}$ and *Cg*MCA-I$^{Mg}$ cation binding sites. a, c**. Positions of Ca$^{2+}$ in *yellow* and the coordinating water molecule in *red* (*Cg*MCA-I$^{Ca}$) respective to Sm$^{3+}$ in *green* (*Tb*MCA-Ib, PDB code: 4AFP) (**a**) and Mg$^{2+}_A$ in *purple* (*Cg*MCA-I$^{Mg}$) (**b**). Loop Gln$^{185}$-Asp$^{199}$ surrounding the Ca$^{2+}$ (*Cg*MCA-I$^{Ca}$) is shown in *green* (**a, c**), the disordered loop Cys$^{162}$-Asp$^{173}$ in *Tb*MCA-Ib is shown in *cyan* (**a**) and loop Asp$^{188}$-Gly$^{196}$ in *Cg*MCA-I$^{Mg}$ is shown in *gray* (**c**). Schematic representation of the coordinated Sm$^{3+}$ (**b**) and Mg$^{2+}_A$ (**d**) as observed in the structures of *Tb*MCA-Ib and *Cg*MCA-I$^{Mg}$, respectively.

suggests that the change in $k_{cat}$ is rather due to the local spatial reorganisation of the catalytic site residues. These observations are supported by the afore mentioned crystal structures of *Cg*MCA-I$^{Ca}$ and *Cg*MCA-I$^{Mg}$ which highlighted that the presence of calcium narrows the distance between the two catalytic residues stabilizing in the same time this conformation (Fig. 4).

In order to follow the maturation process, purified enzyme was stored at 22 °C in the absence of calcium and activity was determined daily in the presence of 10 mM Ca$^{2+}$ over 10 days (Fig. 6). The activity curve is a sigmoid that can be interpreted as follows.

From days 0 to 4, the activity is slightly higher than the background signal ($\sim 10.10^{-6}$ μM min$^{-1}$). The main metacaspase forms E$_0$, E$_1$, E$_2$ identified by gel electrophoresis at the early stage of maturation, are not fully catalytically active (Fig. 1). At day 5, the activity increases sharply reaching a maximum of $\sim 60.10^{-6}$ μM min$^{-1}$ at day 8. This is consistent with the 6-fold increase of $k_{cat}$ observed in the previous experiment. Of these maturation steps, *Cg*MCA-I form E$_3$ identified by gel electrophoresis appears to be the fully active enzyme state. After day 8, all enzyme molecules are mostly in their catalytically active (mature) form.

From an activity point of view, the sigmoid shape of this curve suggests that two enzyme populations exist: a first population (namely E$_{low}$) with lowest specific activity comprising E$_0$, E$_1$, and E$_2$ states; and a second population (namely E$_{high}$) with the highest specific activity composed by the E$_3$ state. In both populations, the distinction between enzymes of the same activity is impossible.

The sigmoid shape of the curve suggests a *trans* mechanism, *i.e.* enzymes of the E$_{high}$ population are able to maturate enzyme of the E$_{low}$ population to afford E$_{high}$ enzymes. Several models were evaluated to describe the kinetic mechanism of the transition from the E$_{low}$ to the E$_{high}$ population (Supplementary Tables 5, 6). The most acceptable model is a two-step model where the first transition from the E$_{low}$ to the E$_{high}$ population follows a *cis* mechanism, whereas the second step is a *trans* mechanism in which enzymes of the E$_{high}$ population catalyse the maturation of the E$_{low}$ enzymes into E$_{high}$ enzymes population, similar to an autocatalytic mechanism (Fig. 6a). In order to validate this hypothesis, we mixed a fully maturated enzyme with a catalytic mutant *Cg*MCA-I$^{C238A/H182A}$, unable to maturate (Supplementary Fig. 12b). We observed after a period of 8 days that the catalytic mutant was indeed maturated by the active enzyme present at low concentration.

Apparent kinetic constants of the first step are about 600 times lower than those of the second step (Supplementary Table 8), thereby explaining why, under our experimental conditions, the appearance of the enzyme with high activity follows nearly an OFF-ON behaviour.

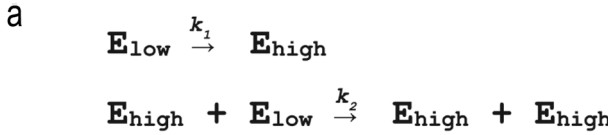

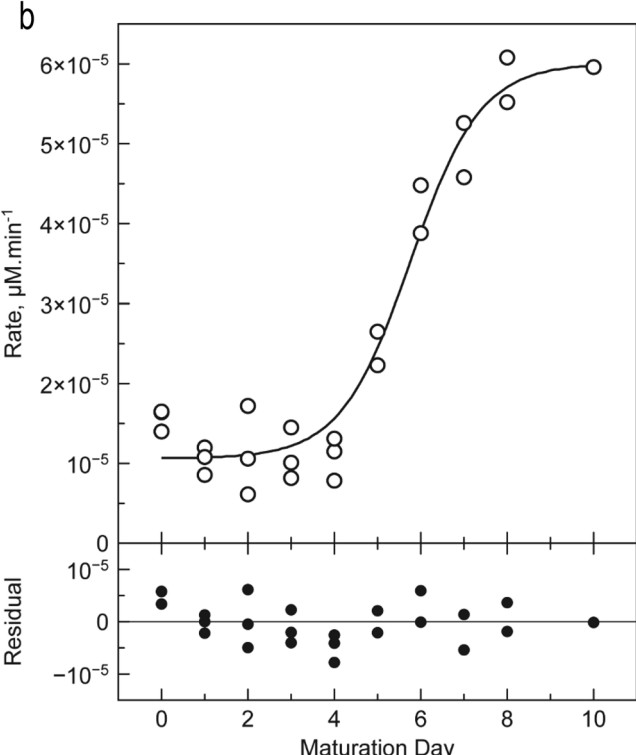

**Fig. 6 Enzymatic characterization of the CgMCA-I maturation process.**
**a** Two-step model describing the transition of enzymes of the $E_{low}$ population into enzymes of the $E_{high}$ population. **b** Hydrolytic activity determined during a 10-days maturation experiment. CgMCA-I (16.5 μM) is stored at 22 °C in 10 mM HEPES buffer pH 7.6, 150 mM NaCl. Reaction conditions: [CgMCA-I] = 2 μM, [Z-GGR-AMC] = 50 μM, 10 mM HEPES buffer pH 7.6, 150 mM NaCl, 10 mM CaCl₂ ($n = 3$ independent experiments).

When the reaction is performed in the presence of EGTA (1 mM) or in the absence of calcium, no significant activity is observed during and after eight days evidencing that calcium is required for the activity. In order to assess the affinity of CgMCA-I for $Ca^{2+}$ at different stages of maturation, we performed nano-DSF experiments and measured the intrinsic fluorescence of CgMCA-I in the absence or presence of $Ca^{2+}$ after different incubation times (0, 1 h, or 24 h) (Supplementary Fig. 13). For CgMCA-I, the apparent $K_d$ is slightly increasing with incubation time, indicating that even the unmatured form of the enzyme is able to bind $Ca^{2+}$ (Supplementary Fig. 13b). Concerning, CgMCA-I at day 8 of maturation, the apparent $K_d$ does not change drastically with incubation time and compared to CgMCA-I at maturation day 0 with 24 h of $Ca^{2+}$ incubation (Supplementary Fig. 13). Altogether, this could indicate that the increased activity from the $E_{low}$ forms to the $E_{high}$ form would be due to the maturation of the enzyme (successive cleavages) rather than to an increased calcium affinity.

Regarding the cleavage site mutants, they all display different behavior from a catalytic point of view (Supplementary Fig. 14). Mutants K263A, K263F, K263W, and R307A all showed a delay

in the activation process. Furthermore, K263F and K263W are less active after eight days of maturation. For mutant R307A, a rapid activation at day 4 but a decreased activity at day 8 was observed, illustrating an aberrant maturation. Mutant R54A showed the most intriguing behavior with a higher activity than that of the wild-type enzyme throughout the maturation process, and could suggest that the N-terminal, when un-mutated, could modulate the metacaspase activity prior to cleavage in the very last step of the maturation. These results also confirmed that the presence of basic residues at the different cleavage sites, at least in vitro, are not mandatory to mature the protein, and that the cleavage is probably more structure-dependent. However, the anarchic levels of activity obtained for the different mutants suggest that a properly matured metacaspase requires the presence of these residues.

**Discussion**
The present study provides functional and structural evidence on the $Ca^{2+}$ dependency of metacaspases for their autocatalytic activity. In CgMCA-I, a complete activation requires at least three cleavages resulting in a fully maturated enzyme with its p20 and p10 domains.

We observed that the first cleavage occurred at the C-terminal of the protein forming an independent domain that can be released from the rest of the protein at denaturing conditions (Fig. 1a, b). The second and third cleavages occurred, respectively, at positions 263 and 307, releasing the central loop CgL6 ($Asp^{264}$-$Arg^{307}$), and eventually at the N-terminal of the protein ($Arg^{54}$) (Fig. 1a, b). In the structure of TbMCA-Ib, the catalytic dyad is buried beneath the N-terminal that encircles the main body of the enzyme, and this conformation could indicate that the cleavage of the N-terminal domain may be a prerequisite for complete activation of the enzyme. Regarding the three-dimensional structures of ScMCA-I or CgMCA-I, the N-terminals are not visible, the structures having been determined using mature enzymes. It was suggested that the N-terminal of ScMCA-I had no auto-inhibition role[16]. The fact that the N-terminal of CgMCA-I is the last part of the enzyme to be cleaved, confirms that its role is not to control or to restrain the autocatalytic activation of this type of enzymes.

Earlier studies reported different catalytic activities regarding the maturation level of metacaspase. For example, processing of Trypanosoma brucei metacaspase 2 TbMCA2 seems to modulate its substrate specificity. The uncleaved/non-processed form (TbMCA2 K55/268G mutant) shows activity towards small synthetic substrates but is unable to cleave larger substrate like azocasein[11]. This could illustrate the fact that CgMCA-I shows a basal activity before maturation, namely the $E_{low}$ forms, prior to achieving a fully maturated and active form, namely the $E_{high}$ form. More surprisingly, Coll et al.[18], showed that, in Arabidopsis thaliana, the function of AtMC2 which is to negatively regulate cell death, does not depend on its catalytic residues[18]. This indicates that the cellular function of metacaspase could go beyond a simple maturation and catalytic activation.

Based on the structure and the identified cleavage sites, we questioned the possibility of an autocatalytic cleavage in cis versus an autocatalytic in trans. As previously indicated for the crystal structure of ScMCA-I[16], the distances between the catalytic center and the cleaved residues are hardly compatible with a cis cleavage. Our studies confirm that the activation of CgMCA-I could rely on a trans activation. If so, this suggests that the specificity of the cleavage is not dependent on the sequence due to the lack of conservation between the different cleavage sites[14–16], but rather depends on the target loops conformation. This could also indicate that in order to avoid systematic cleavage, the active site is only accessible or at least restrained to this extended portion of

the target proteins, with the accessibility ultimately relying on the presence of basic residues. As described previously, the cleavage of the metacaspase could exist in *trans* as observed by Watanabe and Lam for rAtMCP2d[12]. However, the authors suggested that cleavage in *cis* could exist as well in an even more efficient way. Interestingly, this observation is in agreement with our most probable model. Indeed, it may illustrate the fact that transitions from $E_0$ to $E_1$, $E_1$ to $E_2$, and $E_2$ to $E_3$ would be in *cis*. Then, as soon as an $E_3$ form ($E_{high}$ form) appears, the *trans* cleavage would be the most probable. In the cell, we hypothesize that the auto-cleavage could generate an $E_{high}$ form able to transactivate the resting pool of metacaspases.

For substrate accommodation, the active site in metacaspase three-dimensional structures available to date display similar organizations with a loop bundle in which $Cg$L3 and $Cg$L4 are bearing, respectively, the catalytic histidine and cysteine amino acid residues (the loop bearing the catalytic cysteine being termed L2 for the executioner caspase-7[19] and $Sc$MCA-I[16]). The bundle moreover consists of loop $Cg$L7 capping the catalytic site (named L5[19], L3[20] or loop 341[14] for caspase-7, L3 for $Sc$MCA-I[16], and the so-called "280-loop" for $Tb$MCA-I[14]) and loop $Cg$L1 and $Cg$L7α (named L4 for caspase-7[20] and $Sc$MCA-I[16]) on each side of the catalytic site. In this assembly, loop L1 ($Cg$L1) observed in metacaspase structures determined so far shows the very same size and orientation[14,16,19]. As concerns $Cg$L7 (including the αcap) it shows a highly variable sequence, size, and structural organization (Supplementary Fig. 6). In *A. thaliana*, the catalytic site is capped by the linker domain loop L6. This unique conformation forces a lysine (Lys[225]) to enter the catalytic site, resulting in the self-inhibition of $At$MCA-IId[15]. In $Tb$MCA-Ib[14], the structure determined from a calcium soaked crystal shows a complete reorientation of the 280-loop which forms a two-stranded β-sheet with the N-terminal embedded in the catalytic site. In $Cg$MCA-I, two different organizations are observed as well. In the presence of calcium, loop $Cg$L7 is mainly destabilized, whereas in the presence of magnesium, the loop forms a one turn helix which is stabilized by a hydrogen bond between Asn[337] and the side chain of Ser[373] and Gln[374] (Supplementary Fig. 15). $Cg$L7 is further stabilized by the presence of a $Mg^{2+}$ which is coordinated by 6 water molecules interacting with the side chain of Asp[120] and the main chain of Gly[116], Ile[338] and Gly[339] (Supplementary Fig. 15b, c). Under this conformation, $Cg$L7 is entering the catalytic site forming a hydrogen bond between the catalytic His[182] and Thr[331] (Supplementary Fig. 15d, e). In addition, comparative studies between $Cg$MCA-I$^{Ca}$ and $Cg$MCA-I$^{Mg}$ revealed that the presence of the two $Mg^{2+}$ ions in the loop Asn[185]-Asp[199] engenders an increase in the distance between the two catalytic residues. By substituting the magnesium, $Ca^{2+}$ could force the enzyme to switch to an active form and lock it under this configuration. Finally, in the study of $Tb$MCA-Ib, it was suggested that a second site of low affinity[3,21,22] could exist for $Ca^{2+}$ playing a role in the control of the catalytic mechanism, but also that the binding could be dependent on the negatively charged residues in the p10 domain. In the $Cg$MCA-I$^{Mg}$ structure, $Mg^{2+}$ is embedded in a negatively charged pocket formed by residues rather coming from the p20 domain like Asp[120] (Supplementary Fig. 16). We observed three different binding sites for $Mg^{2+}$, one sharing the location of the main $Ca^{2+}$ binding site and the second in the vicinity of the first one, stabilizing the cation binding loop. We hypothesize that the third $Mg^{2+}$ binding site could correspond to the $Ca^{2+}$ low affinity binding site described by McLuskey *et al.*, for $Tb$MCA-Ib[14]. By replacing magnesium, it could reorient loop L7 loop reinforcing the effect of the $Ca^{2+}$ which interacts with the cation binding loop.

Surprisingly, during this study, we observed a very long period of maturation (eight days without calcium and four days in

presence of calcium Fig. 1b). It seems unlikely that a biological mechanism (i.e the autoprocessing of $Cg$MCA-I) that is part of a cell death phenomenon could take place over such a long period of time. In the study of $At$MC-IV[15], the authors found that the plant metacaspase shows complete maturation after 10 min in presence of 12.5 mM $Ca^{2+}$. Interestingly, in a recent publication by Aryamloo et al.[23], it was observed that *C. glabrata*, exposed to different concentrations of caspofungin, exhibited the features of late apoptosis/necrosis after 18 h of incubation. Whereas this does not mean that metacaspase is necessarily involved in this phenomenon, it does indicate at a minimum that the apoptotic phenomenon in *C. glabrata* is relatively short and inferior to one day. Hence, the length of this process in our study could be due to these experiments being performed in vitro and not in vivo. It cannot be excluded that in the cell, protein partners, metacaspase or $Ca^{2+}$ local concentrations or specific cellular localization could accelerate the process.

## Material and methods

**Cloning and site-directed mutagenesis experiments**. A codon-optimized gene encoding the metacaspase from *Candida glabrata* was synthesized by GenScript and cloned into the *Nde*I and *Bam*H1 sites of the pET15b vector (Novagen) with a 6xHis tag fused at the N-terminal ($^{His}Cg$MCA-I). The construct of $Cg$MCA-I with a His-tag at the C-terminal ($Cg$MCA-I$^{His}$) was generated by a first step of amplification using pET15b_$Cg$MCA-I ($^{His}Cg$MCA-I) as template and oligonucleotide primers (Supplementary Table 1) containing *Sal*I and *Sac*I restriction sites. In a second step, the amplified fragment was cloned into *Sal*I and *Sac*I sites of the pET52 vector (Sigma-Aldrich) with a 10xHis tag at the C-terminal. PCR was performed: $1 \times 98\,°C$ for 5 min, $30 \times [98\,°C$ for 30 s; $54\,°C$ for 1 min; $72\,°C$ for 2 min] and $72\,°C$ for 20 min. Cloning steps were performed in a DH5α strain and the construct was verified by DNA sequencing. The characteristics of both constructs are summarized in Supplementary Table 2.

The $Cg$MCA-I gene was mutated with oligonucleotides listed in Supplementary Table 1, according to the principle of PCR-mediated point mutagenesis[24] and was used for all single mutants: C238A, R74A, K283A, R307A. The double-mutant C238A/H182A was obtained using the single mutant C238A as template and the H182A primers. PCR was performed: $1 \times 98\,°C$ for 5 min, $30 \times [98\,°C$ for 30 sec; $50\,°C$ for 1 min; $72\,°C$ for 2 min] and $72\,°C$ for 20 min. All constructs were verified by DNA sequencing.

**Expression and purification of $Cg$MCA-I and mutants**. All constructs of $Cg$MCA-I were overproduced in *Escherichia coli* T7 express strains (New England Biolabs), cultivated in 2 L of Luria-Bertani (LB, Sigma-Aldrich) medium supplemented by $50\,\mu g\,mL^{-1}$ ampicillin at 160 rpm at $37\,°C$. $Cg$MCA-I expression was induced by 0.2 mM isopropyl-β-D-thiogalactopyranoside (IPTG, Sigma-Aldrich) when the culture reached an $OD_{600nm}$ of 0.6; the induction was carried out at 160 rpm and $25\,°C$ for 20 h. Cells were harvested by centrifugation at $5000 \times g$ for 15 min at $4\,°C$ and the pellet was resuspended in 60 mL of 10 mM HEPES pH 7.6, 300 mM NaCl lysis buffer in the absence or the presence of 10 mM $CaCl_2$ and supplemented by one tablet of cOmplete, EDTA free protease inhibitor cocktail (Roche), $25\,\mu g\,mL^{-1}$ lysozyme (Sigma-Aldrich), $5\,\mu g\,mL^{-1}$ DNase I and RNase I (Sigma-Aldrich). Bacteria were lysed by using the Microfluidizer (M110P, Microfluidics) at 1500 bar. The lysate was then centrifuged at $11,000 \times g$ for 45 min at $4\,°C$ and the supernatant was applied to a Co-NTA column (HiTrap® TALON® crude 5 mL, GE Healthcare, UK) beforehand washed with lysis buffer containing 30 mM imidazole (Merck), 10 mM $CaCl_2$ and 1% (*w/v*) glycerol (Euromedex). Elution was

performed with an ÄKTA Purifier (GE Healthcare) with a linear gradient from lysis buffer to 100% of elution buffer containing 500 mM imidazole. The eluted protein was then desalted using an HiPrep© Desalting column (GE Healthcare, UK) equilibrated with a desalting buffer composed of 10 mM HEPES pH 7.6, 150 mM NaCl, in the absence or the presence of 10 mM $CaCl_2$ and 1% (w/v) glycerol prior to the experiment. The eluted protein was concentrated using an Amicon Ultra Centrifugal filter with a 30 kDa molecular weight cut-off (Millipore$^{TM}$ Corporation) and loaded on a gel filtration column (Superdex-200, GE Healthcare) previously equilibrated in the desalting buffer. The protein concentration was assessed with a NanoDrop (Thermo Fisher Scientific) and the purity and molecular mass of the protein were confirmed by SDS-PAGE using a gel composed of 4% stacking and 15% running gel and following Coomassie staining.

**Maturation assays**. Analysis of the maturation steps of *Cg*MCA-I was performed by purifying the protein in the presence or the absence of $CaCl_2$ (10 mM), $MnCl_2$ (10 mM), $MgCl_2$ (10 mM), $ZnCl_2$ (10 mM) or EGTA (1 mM) (Supplementary Table 3) using a BioSprint 96 (Qiagen) for a rapid purification of the His-tagged protein on Ni-NTA-coated magnetic agarose beads. *Cg*MCA-I was overproduced in *E. coli* T7 express strains (New England Biolabs®), cultivated in 200 mL of Luria-Bertani (LB, Sigma-Aldrich) medium supplemented with 50 µg.mL$^{-1}$ ampicillin at 160 rpm at 37 °C. *Cg*MCA-I expression was induced by 0.2 mM isopropyl-β-D-thiogalactopyranoside (IPTG, Sigma-Aldrich) when the culture reached and $OD_{600nm}$ of 0.6; the induction was carried out at 160 rpm and 25 °C for 20 h. Cells were harvested by centrifugation at 5000 × *g* for 15 min at 4 °C and the pellet was re-suspended in 5 mL of the corresponding lysis buffer (listed in Supplementary Table 2) and supplemented by CLAPA 1X (Chymostatin 1 µg mL$^{-1}$, Leupeptin 1 µg mL$^{-1}$, Antipain 1 µg mL$^{-1}$, Pepstatin 1 µg mL$^{-1}$, Aprotinine 5 µg mL$^{-1}$), 25 µg mL$^{-1}$ lysozyme (Sigma-Aldrich), 5 µg mL$^{-1}$ DNase I and RNase I (Sigma-Aldrich). Bacteria were lysed by sonication on ice during 3 × 30 s, using a SONIFIER 250 (Branson). The bacterial lysate was clarified by centrifugation at 11,000 x *g* for 20 min at 4 °C and the supernatant was loaded onto the BiosPrint blocks. The proteins bind *via* their 6xHis tag to the Ni-NTA-coated surface of the magnetic beads that are then efficiently washed with 100 µL of the corresponding buffer (listed in Supplementary Table 3), removing nonspecifically bound protein while leaving the protein on the matrix. Pure protein is eluted in 100 µL of the corresponding elution buffer (listed in Supplementary Table 4). Elution fractions are collected at days 0, 4, and 8 after purification, supplemented with Laemmli buffer and further stored at − 20 °C prior to their analysis by SDS-PAGE 15%. Elution fractions at days 0, 4, and 8 were also further analyzed by western blot after being separated in a 15% SDS-PAGE and transferred to nitrocellulose membranes for immunoblot. The membrane was incubated with a primary monoclonal anti-His antibody (mouse, 1:5000, SIGMA H1029). Immunoblots were detected using phosphatase alkaline-conjugated anti-mouse secondary antibody (1:10,000, SIGMA A3562) and Sigma-Fast phosphatase alkaline substrate (SIGMA, F4648).

## Mass spectrometry

*LC-MS/MS analysis*. Analysis of *Cg*MCA-I fragments was performed to determine the cleavage sites.

A few hundred ng of proteins samples were injected and analyzed using an Ultimate 3000 nano-RSLC (Thermo Scientific, San Jose California) coupled on line with a Q Exactive HF mass spectrometer *via* a nano-electrospray ionization source (Thermo Scientific, San Jose California).

Samples were injected and loaded on a C4 Acclaim PepMap300 trap-column 300 µm ID × 5 mm, 5 µm, 300 Å, (Thermo Scientific) for 3 min at 20 µL min$^{-1}$ with 2% acetonitrile, 0.05% trifluoroacetic acid in $H_2O$ and then separated on a C4 Acclaim Pepmap300 nano-column, 15 cm × 75 µm i.d, 5 µm, 300 Å (Thermo Scientific) with a 30 min linear gradient from 3.2 to 80% buffer B (A: 0.1% formic acid in $H_2O$, B: 100% acetonitrile, 0.1% formic acid) and then from 80 to 95% of B in 1 min, hold for 5 min and returned to the initial conditions in 1 min for 13 min. The total duration was set to 50 min and the flow rate was 300 nL min$^{-1}$. The oven temperature was kept constant at 40 °C.

The mass spectrometer was operated in the positive full MS mode and using a scan range from 500 to 2000 mass-to-charge (m/z). The mass resolution was set to 240,000 at 200 m/z, the AGC target was 3e$^6$, and the maximum injection time was 100 ms.

The deconvolution analyses were processed using Biopharma Finder 2.0 (Thermo Scientific) with the default method Xtract-average over selected mass range (isotopically resolved) that uses a minimal charge state of 5, with at least 3 different charge states to identify a compound, and the accuracy mass tolerance for the deconvoluted MW was set at 20 ppm. The average deconvoluted mass spectrum was obtained resulting in the achievement of a list of MWs corresponding to the detected fragments present in the metacaspase sample. Deconvoluted masses with a relative abundance lower than 1% were filtered out. The resulting list of MWs are compared to the theoretical mass of the amino acid sequence of metacaspase with the potential presence of oxidations on methionines using FindPept tool on the Expasy website (SIB Swiss Institute of Bioinformatics, Expasy).

**Enzyme kinetics and characterization**. The enzyme solution (16.5 µM) was stored in 10 mM HEPES pH 7.6, 150 mM NaCl at 22 °C in the absence of calcium for at least 8 days. Every day, the activity of the enzyme was measured as follows: reactions were performed in 10 mM HEPES pH 7.6, 150 mM NaCl supplemented by 10 mM $CaCl_2$ with 50 µM Z-GGR-AMC as substrate (10 µL). The reaction media incubated at 25 °C before starting the reaction by the addition of 12.5 µL enzyme (2 µM). The reaction volume was 100 µL. The release of AMC was followed every minute by measuring the fluorescence using an excitation wavelength of 355 nm (bandwidth 9 nm) and an emission wavelength of 460 nm (bandwidth 20 nm). Under these conditions the specific response of AMC is 4309 µM$^{-1}$. Kinetics were recorded using a Tecan M200 microtiter plate reader (Salzburg, Austria) using black 96-well plates, and all reactions were performed at least in duplicate. Linear parts of the kinetics are used as initial rates and expressed as µM min$^{-1}$. The control was obtained by replacing the enzyme solution by buffer. Determination of the kinetic constants was done using the same protocol, the only difference being that the final substrate concentration was 0–50 µM and the metacaspase concentration 4 µM. Similar reactions were conducted in the presence of EGTA (1 mM) and in the absence of calcium. Data analysis was performed and kinetic parameters were adjusted using the software package Dynafit[25].

*Script 1* from the Dynafit software package was used for the determination of the kinetic constants at day 0 (Supplementary Table 5). The script takes into consideration the chemical hydrolysis of Z-GGR-AMC in the absence of any protease. Initial parameters were set arbitrary. A similar script was used to determine the constants of the enzyme at day 8.

The rapid-equilibrium hypothesis is used. Concentrations are in µM.

```
[task]
    task = fit;
    data = rate;
    approximation = rapid-equilibrium;

[mechanism]
    E + S <===> ES  :  Km dissoc
    ES --->E + P    :  kcat
    S --->P         :  kc

[constants]
    kcat = 1 ?
    Km = 25 ?
    kc = 1 ?

[responses]
    P = 0.004309

[concentrations]
    E = 0;

[data]
    variable S
    directory …
    sheet MM.csv
    column 2 | concentration E = 4 | label + Enz
    column 3 | concentration E = 0 | label - Enz

[output]
    directory …
[settings]
{Output}
    XAxisLabel = [Z-GGR-AMC], microM
    YAxisLabel = rate, Fluorescence units.min-1
[end]
```

*Model description.* The maturation of $E_{low}$ into $E_{high}$ population could result from three elementary steps:

- a *cis* mechanism where $E_{low}$ is converted into $E_{high}$ following first order kinetics
- a *trans* mechanism in which two $E_{low}$ molecules are converted into at least one $E_{high}$ molecule (the second being $E_{low}$ or $E_{high}$) following second order kinetics
- a *trans* mechanism where $E_{low}$ is converted into $E_{high}$ by a pre-existing $E_{high}$ molecule following second order kinetics. This mechanism assumes that one of the previous mechanisms also occurs.

These mechanisms are not exclusive and others could be envisioned by combination of these three elementary steps. Trans mechanisms are usually associated to cooperativity, autocatalysis, or self-amplification processes leading to a sigmoidal behaviour and appears suitable for describing the metacaspase maturation. In order to elucidate the mechanism, several models were implemented and discriminated using the software package Dynafit (Supplementary Table 6). In all models, only two enzyme populations were considered because model discrimination is performed using one single parameter (*e.g.* activity). Therefore, the early maturation steps such as the conversion of $E_0$ to $E_1$, both

low-activity forms, could not be differentiated. The initial $E_{low}$ concentration is set at $8\,\mu M$ and is not adjusted to the experimental data. The model response is the reaction rate that is directly related to the concentration of $E_{low}$ and $E_{high}$, each form having its own specific response:

$$\text{Rate}\,(\mu M.min^{-1}) = r_{low} \cdot [E_{low}] + r_{high} \cdot [E_{high}] \qquad (1)$$

where $r_{low}$ and $r_{high}$ are the specific response of $E_{low}$ and $E_{high}$, respectively, initially set to $1.25 \times 10^{-6}\,\mu M\,min^{-1}\,\mu M^{-1}$ and to $7.5 \times 10^{-6}\,\mu M\,min^{-1}\,\mu M^{-1}$ and $[E_{low}]$ and $[E_{high}]$ the molar concentrations of $E_{low}$ and $E_{high}$ respectively.

Parameters that can be adjusted are specific responses ($r_{low}$ and $r_{high}$) and kinetic constants ($k_1$ and $k_2$). Model discrimination was performed using three set of initial parameters for each model, leading to 18 different adjustments (Supplementary Table 7). For a detailed explanation regarding the discrimination methodology please refer to the software handbook or to the literature[26,27]. A single model is found to be acceptable according to the coefficient of variation, adjustable parameters, relative sum of squared deviations, differential AIC/BIC criterion, and Akaike/Bayesian weight. The *cis-trans* model using parameter set 3 where the $k_1$ constant is set to $0.001\,day^{-1}$. Optimized parameters are reported in Supplementary Table 8.

**Thermostability and calcium affinity**. The thermostability of *Cg*MCA-I was assessed using nanoscale differential scanning fluorimetry (nanoDSF; NanoTemper) by measuring the intrinsic dual-UV florescence change in tryptophan and tyrosine residues at emission wavelengths of $330\,nm$ and $350\,nm$. Experiments were conducted either with the protein at day 0 and day 8 of maturation at $18\,°C$ without $CaCl_2$. Prior to fluorescence measurements, enzyme samples were supplemented with $CaCl_2$ (final concentrations are 0, 0.1, 0.25, 0.5, 1, 2, 5, 7, 10 and 20 mM) and placed in the DSF capillaries. Measurements were performed immediately, after 1 h or after 24 h incubation at $18\,°C$. Samples were then screened over a temperature range from $25\,°C$ to $95\,°C$ using a heating gradient of $1\,°C/min$. Each experiment was conducted in duplicates.

Melting temperature for both *Cg*MCA-I at maturation day 0 and day 8 without, after 1 h and after 24 h of maturation are respectively summarized in Supplementary Tables 9 and 10.

Delta Tm were calculated from the difference between melting temperatures in the presence of each $CaCl_2$ concentration and the melting temperature without any $CaCl_2$ ($Tm_0$) for each experiment (Supplementary Fig. 13a). The apparent $K_d$ was determined by curve-fitting using a single binding site model for calcium cations (Supplementary Fig. 13b).

$$\triangle Tm = \frac{\triangle_{Tm}^{max} \cdot [Ca^{2+}]}{K_d + [Ca^{2+}]} \qquad (2)$$

**Crystallization, data collection, and structure determination**. Screening of crystallization conditions was carried out using the sitting-drop vapor-diffusion method employing commercial crystallization kits: MB Class suite (Qiagen), MB Class II (Qiagen), PEG I (Qiagen), PEG II (Qiagen), PEG ION (Hampton Research), Crystal screen (Hampton Research), MPD (Qiagen), MIDAS+ (Molecular Dimensions) and JCSG+ suite (Qiagen). MRC crystallization plates (Molecular Dimensions) were filled using a Mosquito® crystallization robot (SPT Labtech) and two protein/crystallization agent ratios were tested ($200\,nL + 200\,nL$ and $200\,nL + 100\,nL$ drops equilibrated against $70\,\mu L$ in the reservoir). Two batches of *Cg*MCA-I led to crystal growth. A first preparation of *Cg*MCA-I at maturation

day 0, concentrated to 10.3 mg mL$^{-1}$ in 10 mM HEPES pH 7.6, 150 mM NaCl, 10 mM CaCl$_2$ resulted in crystals of the protein complexed with Ca$^{2+}$ (CgMCA-I$^{Ca}$). Crystals of CgMCA-I$^{Ca}$ complexed with Ca$^{2+}$ were grown in 0.2 M (NH$_4$)$_2$SO$_4$, 30% (w/v) PEG 4000. A second preparation of CgMCA-I at maturation day 4, concentrated to 8.2 mg mL$^{-1}$ in 10 mM HEPES pH 7.6, 150 mM NaCl, 10 mM CaCl$_2$ led to crystal growth of CgMCA-I$^{Mg}$ complexed with Mg$^{2+}$ in 0.2 M Mg Acetate, 20% (w/v) PEG 3350. Crystals appeared in both conditions after three months. Crystals were harvested and further cryo-protected in the same solution to which 15% ethylene glycol had been added. X-ray diffraction data were collected at PXIII beamline (SLS, Suisse) at wavelengths of 0.99999 Å for CgMCA-I$^{Ca}$ data and at PROX-IMA1 (SOLEIL, Paris) at 0.97857 Å for CgMCA-I$^{Mg}$ data. Data were indexed, integrated, and scaled using programs from the XDS package[28]. The phase problem was solved by molecular replacement using phenix.phaser (PHENIX[29]) with the structure of *Saccharomyces cerevisiae* metacaspase (ScMCA-I, PDB entry 4F6O) as search model (66,5% identity)[30]. Model building was performed using Coot[31] and refinement was carried out using PHENIX[29]. Data collection statistics for CgMCA-I$^{Ca}$ and CgMCA-I$^{Mg}$ are shown in Table 1.

**Small-Angle X-ray Scattering**. Samples purified by gel filtration (Superdex-200, GE Healthcare) in 10 mM HEPES pH 7.6, 150 mM NaCl, 1% glycerol, 10 mM CaCl$_2$ were concentrated to 9.6 mg mL$^{-1}$. SAXS data were collected at the SWING beamline, SOLEIL (Paris-Saclay) in a SEC coupled mode using an analytical Superdex S200 Increase 5/150 (GE Healthcare Life Sciences, U.K) equilibrated in 10 mM HEPES pH 7.6, NaCl 300 mM, 1% glycerol, 10 mM CaCl$_2$. Initial processing was done using PRIMUS[32] and calculation of P(r) was carried out using CRYSOL[33]. Ab initio envelope models of CgMCA-I were generated from the experimental data using DAMMIF[34]. The envelopes were superimposed onto the crystal structure using the CHIMERA software (University of California, San Francisco).

**Statistics and reproducibility**. For any activity and interaction analysis (i.e. kinetic constants determination, activity measurements, and thermophoresis), the experiment was conducted using at least two biological and three technical replicates. Representative data of a biological experiment are presented as the mean of two (thermophoresis, $n = 2$) or three (enzymatic assays, $n = 3$, ±SD replicates).

**Figure rendering**. Figures of three-dimensional structures were drawn with PyMol (Schrödinger, http://pymol.org).

**Reporting Summary**. Further information on research design is available in the Nature Research Reporting Summary linked to this article.

## Data availability
The coordinates and structure of CgMCA-I$^{Ca}$ and CgMCA-I$^{Mg}$ have been deposited in the PDB with respective accession numbers 7QP1 and 7QP0. All other relevant manuscript data are available in the source data provided with this paper and from the corresponding authors upon reasonable request. The source data for all plots are shown in Supplementary Data 1. The uncropped and unedited gel images are included in Supplementary Data 2. The SAXS data have been deposited in the SASBDB with accession number SASDQN5.

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

## Acknowledgements

The authors thank the SOLEIL synchrotron for access to beamline PROXIMA1, the European Synchrotron Radiation Facility for access to MX-beamlines and the Swiss Light Source for access to beamline PXIII. Technical support from all the beamlines staff is gratefully acknowledged. Authors acknowledge the contribution of SFR Biosciences (UAR3444/CNRS, US8/Inserm, ENS de Lyon, UCBL), especially V. Gueguen-Chaignon from the Protein Science Facility. Support from the CNRS is acknowledged. SP is supported by the EDCTP2 programme from the European Union grant number RIA2017T-2018 WANECAM-2.

## Author contributions

Conceptualization S.P., A.L.B., N.A; LB; formal analysis, A.P, E.D., C.D., B.D., L.C., N.A.; S.V., L.B., funding acquisition N.A.; investigation L.C., FG, L.B., S.V., supervision., N.A, L.B; validation S.P., A.B., B.D., L.C., L.B., S.V.; writing B.D., L.C., S.V., N.A and L.B.

## Competing interests

The authors declare no competing interests.
