## [Peer Review File · Communications Biology]

Reviewers' comments:

Reviewer #1 (Remarks to the Author):

Léa Conchou and colleagues present the "Structural and molecular determinants of *Candida glabrata* metacaspase maturation and activation by calcium".

Metacaspases play critical roles in the regulated cell death and beyond. This enzyme class shares a high degree of structural homology with caspases, but also exhibit characteristic differences, such as the preferential cleavage after basic P1 residues and the lack of dimerisation accompanying the activation. The latter property is reminiscent to legumains. The activity regulation of metacaspases is as yet insufficiently understood, in particular the role of Ca²⁺ in these processes.

To address these questions, the authors determined crystal structures of the yeast metacaspase type from *Candida glabrata*, CbMCA-I.

1) Materials & methods section, maturation assays: solubility of ZnCl₂ will be much lower than 10 mM at near neutral pH, should be more μM

2) Table 1:

unit cell parameter for Cg Mca1-Mg²⁺ read "957 83.6 87.83". The a unit cell parameter should probably be 95.7. Also, unit cell parameters should have a consistent, and sensible, precision. This is most likely 1 digit after the decimal point. Here it is all mixed up. CC1/2 values (and I / sigma) indicate that high resolution data were cut off to early.

3) coordinates and x-ray data appear not to be deposited with pdb. Neither are entry codes given in the manuscript nor can I find the structures in the (prerelease area of the) pdb. This is really disappointing.

4) SAXS.

It is a pity that the measurements were conducted at one concentration only, in the absence of calcium.

5) calcium affinity. I cannot see that the authors have investigated the affinity of the calcium site(s), e.g. dissociation constants K_d and / or activity modulation effect EC₅₀. It is also unclear whether the different enzymatic reaction intermediates E0 through E3 have different metal affinity. Finally, to understand the physiologic relevance of metal binding it would be interesting to see how /whether the affinities compare /match with the physiologic calcium /magnesium concentration.

6) two-step Elow to Ehigh transition model (cf Figure 6). To test the The activation should be starkly enhanced by spiking in Ehigh species, i.e. E3. Does this species trans-activate E0 directly to E3, without the need for E1 and E2 intermediates?

7) Given the title of the manuscript ("maturation and activation") I would expect more information about the role of the three cleavages after R54, K263, R307. The cleavage after R54, expected to happen last based on SDS PAGE analysis, releases the N-term peptide. It is stated that this peptide has no inhibitory role (line 307-309). While I agree that this conclusion is plausible by the structure, it could be validated by mutating the cleavage site. The cleavages at K263 and R307 release an inserting activation peptide, which is thought to be critical for activation. However, several questions remain:

a) are both cleavages important to gain full activity?

b) does the sequence of cleavages matter, i.e. is the cleavage at R307 necessary for the second cleavage at K263 to occur?

c) Is cleavage a prerequisite for metal binding – or rather the other way, metal binding a prerequisite for cleavage?

The sigmoid activity gain curve in figure 6 could indicate a synergy of both effects.

Minor comments / suggestions

8) Figure 3: I recommend to label the water molecule with H₂O rather than W.

9) typo. "As concerns CgL7 (including the acap) it shows a most variable sequence, size or structural organization (Sup. Figure 3)". → should be sup Fig 2

10) chemical hydrolysis. (for example suppl fig 7). The reader would benefit it were be explicitly said that the chemical hydrolysis occurs is observed in the absence of a protease (which is what I assume).

Reviewer #2 (Remarks to the Author):

In my opinion the article titled "Structural and molecular determinants of *Candida glabrata* metacaspase maturation and activation by calcium" by Conchou et al., is very interesting with original data and very important for future yeast metacaspase studies. In my opinion the data are very conclusive and the text are good, I just make two little appointments that's easy to be modified, and suggest that this manuscript be publish in Communications Biology.

- In page 3, line 70 the authors says that "We identified for the first time a calcium binding site", I disagree cause McLuskey et al., they described a presence of this calcium binding site on TbMCA-Ib structure and the corresponding residues are conserved at CgMCA-I.
- The legend of figure 4 must be rewritten, because it is necessary to spend a lot of time to understand which structure is from CgMCA-ICa and which is from CgMCA-IMg. I suggest that the figure be redone by differentiating the 2 enzymes in a single color to identify them.

Reviewer #3 (Remarks to the Author):

The authors Conchou et al describe maturation of metacaspase CgMCA-I from *Candida globurata* and the role calcium 2+ ions play in it. Unfortunately, I did not enjoy reading this manuscript. On the one hand it is loaded with technical details and data, on the other hand it lacks description and explanation of certain technical details and data, which raise concerns about technical soundness of the work and correctness of conclusions. In addition this manuscript lacks perspective of biological importance of the work and a related conclusion. I hope that my concerns are only the problem of presentation and which the authors will be able to address them in the revised version.

The Ca and Mg crystal structures of CgMCA-I structures were determined at various resolutions. Table 1 provides cell constants at various precision, with cell length of a space group C2 without a decimal point. The Ca bound form contains 4 ligand atoms and 7 solvent molecules which all have lower average B-factors than protein atoms. yet it does not explain what they were. Are there four Ca²⁺ ions? The Mg from diffracted to 1.6Å resolution, it has 36 ligand atoms or ions, yet their average B-factor is 50.2, which is twice as much as average protein B-factor and higher than average B-factor of solvents. Are ligands and ions indeed what the authors claim to be, is their occupancy lower than 1.0? The later is a red flag which raises concern about the correctness as interpretation of ions constitutes some of the core arguments of the conclusions. The PDB validation report, though included, is suggestive, but the authors should address this issues in the manuscript. Maybe even to re-refine the structure. Comparison of B-factors of ions with their surroundings in a crystal diffracting to 1.6Å should provide an unambiguous answer about the number of electrons of each ion. Please elaborate and show their electron density maps.

The authors describe details about fitting of parts of the model to electron density maps, yet they show no maps in the figures 2C and B, E,F, and A (lines 136, 151, 170).

The use of activation and maturation mechanisms is confusing (line 157). I suggest that the authors choose either maturation or activation, not both, otherwise they have to explain the difference between the two mechanisms. I prefer activation.

Then there is the issue of the number of steps in the activation mechanism. The authors observed 3 cleavages and a Ca²⁺ ion dependent step, which makes 4 steps and not 2. It is possible that there are additional steps associated with separation of N- or C-terminal domains following the cleavage, hence potentially there is room for a quite complicated mechanism likely taking place in the first 2 to 4 days unobserved in Figure 6? What is the biological relevance of an 8 days long activation process of an enzyme? Such long periods are usually associated with transformation of the structure such as fibril formation and not with conformational change of a loop? Where are the controls of stability of components after 8 days of measurement? A similar publication on MCA-IV revealed Ca dependent activation mechanism which lasted minutes (Zhu et al., 2020, Nat. Comm.) Hence, a much better explanation of activation measurements and activation mechanism than provided is required to satisfy my concerns.

The supplement contains no references. I believe that this is a glitch of the publisher's processing system as we had the same problem a while ago.

Reviewer #4 (Remarks to the Author):

The authors report the first study of a type I metacaspase in the active form with bound calcium ions. Surprisingly, structures were obtained also with magnesium ions in it. Below I summarized my comments. I think this is an interesting structure, but the manuscript itself needs some rewriting and reformatting of the figures.

Major:

1. Cloning and purification of the WT protease

What construct was used for mutagenesis, CgMCA-I(His) or (His)CrMCA-I? Additionally, it is not clear what the expected size of MCA is. From the affinity chromatography elutions it is not clear which peak corresponds to the full-length size protein. Please, mark it with an arrow.

Interestingly, both prominent bands disappear from the SDS-PAGE gel after SEC. It is very hard to understand how the protein, eluted from the SEC could be used for maturation studies (Fig. 1), since it is not in full-length anymore. Was protein differently prepared and eluted for this experiment? If yes, this should be made clearer in the main text.

2. Presentation of the structures

I understand that authors wanted to show the structures as they are in the asymmetric unit, but it makes no sense for a common reader. Rather move those to the supplement and in Fig. 2 show monomeric units of CgMCA-ICa and CgMCA-IMg in the same orientation for the comparison. Also in the text, it would be good to start with describing what is similar and then proceeding to what is different.

3. General presentation of results

The paper starts with metals, continues with structures and goes back to maturation. I think the manuscript requires rewriting, to start with the maturation (so that the reader understands well what is happening to the protease while it "matures"-here it is crucial to define what this term means.

4. Effect of calcium

In the case of R54A mutation, in line 230 the authors write that the process is stopped before the last step. But from the Fig. 1 I guess one could speculate what does not happen. This should be

elaborated. In the case of R263A mutation, yes, the protein is degraded, but in a different manner than the WT. This should at least be discussed. The results obtained by K263D and K263F mutants are very intriguing. What would be needed here are activity measurements using fluorogenic substrates.

5. Activity

E3 can not be the active form. It is well known that p10 domain is needed for the activity, that is why it is also present in the structure. Please re-label the Fig. 1 and re-write the enzymatic maturation claims. Cleavage does not release the p10 domain, but most probably rearranges it.

6. The 280 loop

It should be discussed, that in TbMCA-Ib, the 280-loop moves towards and not away from the active site in the presence of calcium, so it can not be active as a capping loop as suggested here (lines 324-326).

7. Cleavages

I do not understand where does the first cleavage occur, since in line 299 the authors claim that the p10 domain "is released"; but there is no cleavage marked and the cleavages denoted as the ones between the p20 and p10 domains are "the second and the third" cleavage. Fig. 1 should be merged with Fig. S6 as it is very important. Additionally, I would suggest making the R307 mutant; again, complementing all the mutant characterisation with the activity assays.

8. Putting this work in the wider context:

- activation in trans should be discussed with similar observations made by Watanabe and Lam (2005)
- the importance of the p280 loop (L7 loop) as a calcium binding site by Klemenčič and Funk (2018)
- the unobserved cleavage between the p20 and p10 domains in vivo by Coll et al., 2010

Minor:

Line 87: what is meant by complete maturation?

p20 and p10 are sometimes written as P20 and P10, please unify.

Line 136: change Asn360 to Asn260.

Line 162: please write which MCA (AtMCA-IIa).

Line 228: departure of the N-terminal?

Line 255: this is not a caspase.

Reviewing “Structural and molecular determinants of *Candida glabrata* metacaspase maturation and activation by calcium” – Communication Biology

Replies to Reviewer 1:

Question 1: Materials & methods section, maturation assays: solubility of ZnCl₂ will be much lower than 10 mM at near neutral pH, should be more μM

We agree with the reviewer that ZnCl₂ should be in the μM range when using a pH around 7.5. Initially, we wanted to reproduce an experiment comparable to that carried out in the publication by Wong et al. (2012), in which the authors used a concentration of 10 mM ZnCl₂ at pH 8.0. In the condition used by these authors, the concentration of Zn²⁺ is extremely low as well. According to the literature (Reichl et al, 1975), at pH 7.5, the solubility of zinc is 325 μM mainly as the ions Zn²⁺ (70 %) and Zn(OH)⁺ (30 %) at 25 °C (see Figure 1 of the publication).

Question 2: Unit cell parameter for CgMCA-I^{Mg} read “957 83.6 87.83”. The a unit cell parameter should probably be 95.7. Also, unit cell parameters should have a consistent, and sensible, precision. This is most likely 1 digit after the decimal point. Here it is all mixed up.

We apologize for this error. Unit cell parameters have been corrected to “97.4 97.4 54.6” and to “95.7, 83.6, 87.8” for CgMCA-I^{Ca} and CgMCA-I^{Mg}, respectively, in Table 1 in the manuscript.

CC1/2 values (and I / sigma) indicate that high resolution data were cut off too early.

We agree that the CC1/2 and the I/sigma(I) could indicate an inappropriate data reduction. In order to answer this question, we went back to look at the data and hereunder the XSCALE files for both structures that the reviewer will find hereafter (Table 1, 2). The referee will see that we tried to find the best compromise between CC1/2, I/sigma(I) and R_{meas}. Regarding the structure of CgMCA^{Ca} we reprocessed the data to 2.5 Ångströms (Table 1) and observed that the R-factor and R_{meas} increase considerably while I/sigma(I) and CC1/2 are very low at resolutions higher than 3.0 Å. Moreover, when we use the data at 2.5 Å compared to at 2.8 Å,

the quality of the electron density map is definitely not improved. Considering this, cutting the data at 3 Å seems to us a *bona fide* option.

SUBSET OF INTENSITY DATA WITH SIGNAL/NOISE >= -3.0 AS FUNCTION OF RESOLUTION													
RESOLUTION LIMIT	NUMBER OF REFLECTIONS			COMPLETENESS OF DATA	R-FACTOR observed	R-FACTOR expected	COMPARED	I/SIGMA	R-meas	CC(1/2)	Anomal Corr	SigAno	Nano
40.00	32	4	5	80.0%	4.8%	3.1%	32	68.33	5.2%	99.9*	-38	0.870	4
35.00	13	2	2	100.0%	3.0%	3.3%	13	57.50	3.3%	0.0	-8	0.582	2
20.00	294	31	31	100.0%	2.9%	3.1%	294	69.52	3.1%	99.9*	-23	0.783	31
15.00	626	59	59	100.0%	3.1%	3.2%	626	71.79	3.3%	99.9*	3	0.799	59
10.00	2488	218	218	100.0%	2.9%	3.3%	2488	68.44	3.1%	100.0*	22	1.024	218
6.00	12200	1136	1136	100.0%	3.9%	4.1%	12200	48.08	4.1%	100.0*	17*	0.959	1136
3.20	85671	8101	8101	100.0%	11.6%	11.4%	85671	19.15	12.1%	99.8*	6	0.848	8094
3.00	21901	2039	2040	100.0%	96.2%	97.8%	21901	2.59	101.0%	83.0*	1	0.742	2039
2.80	26600	2660	2661	100.0%	177.1%	178.7%	26600	1.33	186.6%	56.3*	3	0.681	2659
2.50	61631	5769	5776	99.9%	392.8%	400.2%	61631	0.58	412.6%	22.6*	1	0.613	5758
total	211456	20019	20029	100.0%	15.0%	15.2%	211456	12.17	15.8%	99.9*	4	0.755	20000

Table 1. XSCALE data from 2.5 to 40 Å for CgMCA^{Ca} data.

For the CgMCA^{Mg} structure, we reprocessed the data to 1.53 Å resolution (Table 2). Here again, the R-factor and Rmeas values are multiplied by two with an I/sigma(I) ratio being inferior to 1. Moreover, the CC1/2 value is clearly too low at this higher resolution cut-off (44.7). We therefore sincerely believe that cutting the data at 1.6 Å is the best choice.

However, after having looked at the parameters shown in these two tables, if the reviewer does not agree, we would be happy to reconsider the processing of the data.

SUBSET OF INTENSITY DATA WITH SIGNAL/NOISE >= -3.0 AS FUNCTION OF RESOLUTION													
RESOLUTION LIMIT	NUMBER OF REFLECTIONS			COMPLETENESS OF DATA	R-FACTOR observed	R-FACTOR expected	COMPARED	I/SIGMA	R-meas	CC(1/2)	Anomal Corr	SigAno	Nano
40.00	4	3	8	37.5%	29.2%	20.0%	2	6.69	41.3%	0.0	0	0.000	0
30.00	14	6	7	85.7%	4.8%	3.9%	14	19.63	6.0%	100.0*	0	0.000	0
20.00	101	39	40	97.5%	4.4%	4.1%	96	19.65	5.4%	99.9*	71	1.498	6
15.00	201	64	64	100.0%	3.6%	4.2%	198	20.48	4.4%	99.8*	13	0.934	26
10.00	919	267	269	99.3%	4.1%	4.2%	917	23.04	4.9%	99.6*	12	0.908	153
6.00	4851	1330	1337	99.5%	4.2%	4.5%	4848	21.62	5.0%	99.7*	-2	0.826	892
3.20	32436	9309	9349	99.6%	4.5%	4.7%	32317	21.47	5.3%	99.7*	-2	0.802	4516
2.50	42761	11999	12035	99.7%	6.9%	6.7%	42666	14.85	8.1%	99.5*	0	0.832	5699
2.10	54093	15654	15700	99.7%	11.5%	11.0%	53872	9.86	13.7%	98.8*	-1	0.825	6217
1.80	79575	22551	22647	99.6%	24.9%	24.8%	79167	5.25	29.3%	96.5*	1	0.803	9720
1.60	88386	25833	25907	99.7%	72.4%	80.8%	87922	1.85	86.0%	76.1*	-1	0.714	9404
1.53	43509	12461	12504	99.7%	126.7%	151.4%	43143	0.94	149.3%	44.7*	-1	0.681	5256
total	346850	99516	99867	99.6%	9.2%	9.5%	345162	7.51	10.9%	99.7*	0	0.776	41884

Table 2. XSCALE data from 1.53 to 40 Å for CgMCA^{Mg} data.

Question 3: Coordinates and x-ray data appear not to be deposited with pdb. Neither are entry codes given in the manuscript nor can I find the structures in the (prerelease are of the) pdb. This is really disappointing.

We are sorry that the structures are not yet available but currently on hold until publication. The structure deposition for the two structures was conducted in parallel with the submission of the manuscript. CgMCA^{Ca} has the PDB entry code 7QP1 and CgMCA^{Mg} the PDB entry code 7QP0. This information is now mentioned in the manuscript, line 162-163, page 6. Furthermore,

the validation reports have been sent to Communication Biology the 2022-01-04. We hope that they are now available.

Question 4: SAXS. It is a pity that the measurements were conducted at one concentration only, in the absence of calcium.

We apologize for this. As mentioned in the supplementary information, the SEC-SAXS experiment that we referred to in the manuscript was conducted in the presence of Ca^{2+} (“Samples purified by gel filtration (Superdex-200, GE Healthcare) in 10mM HEPES pH 7.6, 150 mM NaCl, 1% glycerol, 10 mM CaCl_2 were concentrated to 9.6 mg.mL^{-1} ”, page 12). In addition, it has been corrected in the **Material and methods** section where it had been omitted. These data allowed us to superpose the CgMCA^{Ca} structure onto the SAXS envelope.

Another SEC-SAXS experiment was also conducted in the presence of Ca^{2+} (same buffer conditions as described above) with a sample concentrated to 10.6 mg.mL^{-1} (Figure 1). As observed for the first SEC-SAXS experiment, $I(q)$ values at small angles indicate almost no aggregation confirming the monodispersity of the sample. Moreover, both SEC-SAXS experiments were conducted with sample in a concentration range similar to what has been used for the crystallographic studies and corresponding to the maximum concentration before aggregation occurs (cf. Material and methods, lines 487 and 490, page 16). Actually, it seemed to us that we were at optimal conditions, with no aggregation or repulsion issues and with a maximal concentration giving us the best resolution. Furthermore, as we did not perform batch experiments but only SEC-SAXS experiments, we did not use a number of concentrations since we were already at very good experimental conditions for this study.

Figure 1. Small Angle X-ray Scattering studies of CgMCA-I. **A.** *Ab initio* model of CgMCA-I generated from the SEC-SAXS experimental data using the program DAMMIF³. The crystal structure of CgMCA-I^{Ca} was superimposed onto the calculated (18 Å) SAXS envelope using the software CHIMERA (University of California, San Francisco) and confirmed a monomeric structure of the enzyme in solution. **B.** Log I(q) *versus* q was generated using PRIMUS⁴. **C.** P(r) *versus* r profile from the data in (B) was carried out using PRIMUS⁴. **D.** SAXS data for CgMCA-I.

Question 5: Calcium affinity. I cannot see that the authors have investigated the affinity of the calcium site(s), e.g. dissociation constants K_d and / or activity modulation effect EC₅₀. It is also unclear whether the different enzymatic reaction intermediates E₀ through E₃ have different metal affinity.

Based on the referees suggestion, we performed calcium binding affinity experiments on CgMCA-I. For this purpose, we used nanoscale differential scanning fluorimetry (nanoDSF) to determine melting points of the protein by measuring the intrinsic dual-UV fluorescence change in tryptophan and tyrosine residues at emission wavelengths of 330 nm and 350 nm (cf. Material

and methods, “**Thermostability and calcium affinity**”, lines 472-480, pages 15-16). This experiment was conducted at 18 °C without CaCl₂, with the protein at day 0 and at day 8 of maturation. Prior to fluorescence measurements, enzyme samples were supplemented with CaCl₂ (final concentrations are 0, 0.1, 0.25, 0.5, 1, 2, 5, 7, 10 and 20 mM) and placed in the DSF capillaries. Measurements were performed immediately, after 1 h or after 24 h incubation at 18 °C. Samples were then screened over a temperature range from 25 °C to 95 °C using a heating gradient of 1 °C/min. Each experiment was conducted in duplicate.

In the absence of calcium, regarding CgMCA-I maturation at day 0, the melting temperature (T_{m0}) is around 45 °C without or after 1 hour incubation and increases to 47 °C after 24 hours of incubation (Table 3). At maturation day 8, the T_{m0} value is higher (around 49 °C) and hardly fluctuates at different incubation times (Table 4).

For maturation day 0 and 8, we could observe that the melting temperature increases with the CaCl₂ concentration and reaches a maximum value of 64-66 °C from 5 mM of CaCl₂ independent of the incubation time (Table 3, Table 4). From these results, it can be observed that the presence of CaCl₂ seems to cause a thermodynamic stabilization of the enzyme. On the other hand, it seems that the maturation process itself allows the stabilization of the protein, resulting from a less important variation of the melting temperature at maturation day 8 between T_{m0} and T_{m20} (melting temperature with 20 mM of CaCl₂) regardless of the incubation time.

CgMCA-I D0											
Without incubation				1 hour incubation				24 hours incubation			
Protein (mg/mL)	Ligand (mM)	T _m (°C)	Delta T _m (°C)	Protein (mg/mL)	Ligand (mM)	T _m (°C)	Delta T _m (°C)	Protein (mg/mL)	Ligand (mM)	T _m (°C)	Delta T _m (°C)
1	0	45	-0.05	1	0	44.9	-0.05	1	0	47.2	0.05
1	0	45.1	0.05	1	0	45.1	0.05	1	0	47.1	-0.05
1	0.1	48.5	3.45	1	0.1	49.3	4.3	1	0.1	54.1	6.95
1	0.1	48.8	3.75	1	0.1	49.2	4.2	1	0.1	54	6.85
1	0.25	53.6	8.55	1	0.25	54	9	1	0.25	58.1	10.95
1	0.25	53.7	8.65	1	0.25	53.9	8.9	1	0.25	58.2	11.05
1	0.5	58.4	13.35	1	0.5	58.7	13.7	1	0.5	59.6	12.45
1	0.5	58.3	13.25	1	0.5	58.7	13.7	1	0.5	59.7	12.55
1	1	59.8	14.75	1	1	60	15	1	1	61.1	13.95
1	1	59.8	14.75	1	1	60.1	15.1	1	1	61	13.85
1	2	60.9	15.85	1	2	61.4	16.4	1	2	63.1	15.95
1	2	60.9	15.85	1	2	61.4	16.4	1	2	63.1	15.95
1	5	63.4	18.35	1	5	63.6	18.6	1	5	65	17.85
1	5	63.5	18.45	1	5	63.6	18.6	1	5	65	17.85
1	7	63.8	18.75	1	7	64.1	19.1	1	7	65.4	18.25
1	7	63.8	18.75	1	7	64.1	19.1	1	7	65.5	18.35
1	10	64.2	19.15	1	10	64.4	19.4	1	10	65.7	18.55
1	10	64.2	19.15	1	10	64.4	19.4	1	10	65.8	18.65
1	20	64.6	19.55	1	20	64.9	19.9	1	20	66.1	18.95
1	20	64.7	19.65	1	20	64.9	19.9	1	20	66.1	18.95

Table 3. Melting temperature (T_m) of CgMCA-I, at maturation day 0 (D0), at various CaCl₂ concentrations without and with incubation. Delta T_m values are calculated from the

average T_{m0} (T_m without any ligand) for all three conditions, without and at 1 hour or at 24 hours of incubation at different CaCl_2 concentrations prior to fluorescence measurements.

Cg MCA-I D8											
Without incubation				1 hour incubation				24 hours incubation			
Protein (mg/mL)	Ligand (mM)	T_m (°C)	Delta T_m (°C)	Protein (mg/mL)	Ligand (mM)	T_m (°C)	Delta T_m (°C)	Protein (mg/mL)	Ligand (mM)	T_m (°C)	Delta T_m (°C)
1	0	48.8	0	1	0	49.2	0.1	1	0	49.2	0
1	0	48.8	0	1	0	49	-0.1	1	0	49.2	0
1	0.1	53.8	5	1	0.1	54.8	5.7	1	0.1	54	4.8
1	0.1	53.8	5	1	0.1	54.5	5.4	1	0.1	54.1	4.9
1	0.25	57.8	9	1	0.25	58.3	9.2	1	0.25	58.4	9.2
1	0.25	58.1	9.3	1	0.25	58.1	9	1	0.25	58.4	9.2
1	0.5	60.1	11.3	1	0.5	60.3	11.2	1	0.5	60.2	11
1	0.5	60.1	11.3	1	0.5	60.3	11.2	1	0.5	60.4	11.2
1	1	61.7	12.9	1	1	61.7	12.6	1	1	61.9	12.7
1	1	61.7	12.9	1	1	61.7	12.6	1	1	61.9	12.7
1	2	63.1	14.3	1	2	63.3	14.2	1	2	63.2	14
1	2	63.1	14.3	1	2	63.2	14.1	1	2	63.3	14.1
1	5	64.6	15.8	1	5	64.6	15.5	1	5	64.5	15.3
1	5	64.5	15.7	1	5	64.6	15.5	1	5	64.6	15.4
1	7	64.8	16	1	7	64.9	15.8	1	7	64.8	15.6
1	7	64.8	16	1	7	65	15.9	1	7	64.7	15.5
1	10	65	16.2	1	10	65.2	16.1	1	10	64.8	15.6
1	10	65	16.2	1	10	65.2	16.1	1	10	64.9	15.7
1	20	65.3	16.5	1	20	65.5	16.4	1	20	65.2	16
1	20	65.3	16.5	1	20	65.4	16.3	1	20	65.3	16.1

Table 4. Melting temperature (T_m) of CgMCA-I, at maturation day 8 (D8), at various CaCl_2 concentrations without or with incubation. Delta T_m values are calculated from the average T_{m0} (T_m without any ligand) for all three conditions, without and at 1 hour or at 24 hours of incubation at different CaCl_2 concentrations prior to fluorescence measurements.

Delta T_m values were calculated from the difference between melting temperatures in the presence of CaCl_2 at each concentration and the melting temperature without any CaCl_2 (T_{m0}) for each experiment (Figure 2). The apparent K_d was determined by curve-fitting using a single binding site model for calcium cations (Figure 3).

Equation:
$$\Delta T_m = \frac{\Delta T_m^{max} \cdot [Ca^{2+}]}{K_d + [Ca^{2+}]}$$

It can be observed that at day 0 of maturation, the affinity for CaCl_2 is slightly increasing with incubation time in the presence of calcium. K_d values of 0.3231 ± 0.0171 , 0.301 ± 0.0137 and 0.2237 ± 0.0157 mM are determined for experiments without, after 1 hour and after 24 hours of incubation, respectively. Concerning the enzyme at day 8 of maturation, the apparent K_d (values of 0.2287 ± 0.0069 mM, 0.2233 ± 0.009 mM, 0.2221 ± 0.0072 mM, without incubation, at 1 hour and at 24 hours of incubation, respectively) does not change significantly with incubation time, thereby indicating that the protein presents a level of maturation for which the maximum calcium affinity has been reached.

Altogether, this indicates that the enzyme is able to bind calcium even under its unmaturation form. Furthermore, the affinity does not change drastically over time. This could indicate that the increased enzymatic activity from the E_{low} forms to the E_{high} form is more dependent on the enzyme maturation (successive cleavages) rather than on increased calcium affinity.

Figure 2. Delta Tm variation of CgMCA-I, at maturation day 0 (D0) and 8 (D8), in a range of CaCl₂ concentration without or with incubation time. Delta Tm values are calculated from the average Tm₀ (Tm without any ligand) for all three conditions, without and at 1 hour or at 24 hours of incubation with and at different CaCl₂ concentrations prior to fluorescence measurements. All measurements were made in duplicate.

Figure 3. Apparent K_d for CgMCA-I at day 0 (D0) and 8 (D8) of maturation without or at 1h and at 24h of incubation with CaCl₂. Apparent K_d values were determined by curve-fitting using a single binding site model for calcium cations and are indicated with associated errors.

Regarding the binding of magnesium ions, results are more difficult to interpret because no clear saturation plot is obtained and possibly due to the three magnesium ions are observed in the crystal structure. The general behavior is that magnesium ions decrease the T_m of the enzyme, suggesting that the protein is less stable. At magnesium concentrations below 1 mM, a saturation behavior seems to be present but with a ΔT_m being less than 1°C. Determination of K_d is tedious since experimental data obtained, showed $378 \pm 228 \mu\text{M}$ at day 0 and $194 \pm 122 \mu\text{M}$ at day 8 (Figure 4). At higher concentrations, the T_m decreases nearly linearly up to 10 mM suggesting that magnesium destabilizes the enzyme by a mechanism that is not a simple protein-ligand interaction. At higher concentrations, it was not possible to obtain suitable measurements using Differential Scanning Fluorimetry.

Figure 4. Effect of magnesium ions on T_m as determined by nano-DSF. No clear saturation behavior is observed. The insert corresponds to the delta T_m determined at concentrations below 1 mM Mg^{2+} including a tentative of curve fitting using the simplest single site model.

As reported for mammals, the basal cytosolic calcium concentration is maintained around 50 to 200 nM under physiological conditions⁵. Upon apoptotic signal, this level is thought to increase as in mammals to a level of 100 μM ⁵. This is approximately 400 times higher than the K_d measured for the mature form of *CgMCA-I*, leading to the conclusion that upon cell death stimuli, intracellular levels of calcium can trigger the activation and thereby the function of the enzyme.

As opposed hereto, *TbMCA-I* affinity studies also revealed a much higher K_d ($900 \pm 100 \mu M$) for the high affinity calcium binding site, than the basal level of calcium in the parasite's endosomes, where the enzyme resides⁶. The authors suggested that *in vivo* activation of the enzyme will be triggered in response to certain stimuli that will either raise calcium levels or lower the calcium activation threshold⁶.

These results have been added to the manuscript (lines 311-320, page 10-11): “In order to assess the affinity of *CgMCA-I* for Ca^{2+} at different stages of maturation, we performed nano-DSF experiments and measured the intrinsic fluorescence of *CgMCA-I* in absence or presence of Ca^{2+} after different incubation times (0, 1 hour or 24 hours) (Sup. Figure 13). For *CgMCA-I*, the apparent K_d is slightly increasing with incubation time, indicating that even the unmaturred

form of the enzyme is able to bind Ca^{2+} (Sup. Figure 13 B). Concerning, CgMCA-I at day 8 of maturation, the apparent K_d does not change drastically with incubation time and compared to CgMCA-I at maturation day 0 with 24 hours of Ca^{2+} incubation (Sup. Figure 13). Altogether, this could indicate that the increased activity from the E_{low} forms to the E_{high} form would be more dependent on the maturation of the enzyme (successive cleavages) rather than on increased calcium affinity.”

Question 6: Two-step E_{low} to E_{high} transition model (cf. Figure 6). To test the activation should be starkly enhanced by spiking E_{high} species, i.e. E_3 . Does this species trans-activate E_0 directly to E_3 , without the need for E_1 and E_2 intermediates?

From day 0 to day 4, the activity of the enzyme remains low but several forms are observed by SDS-PAGE (E_1 , E_2) (Figure 1 of the manuscript). As indicated in our enzymatic model, all these forms present similar activity towards the substrate Z-GGR-AMC. Based on our maturation hypothesis, these forms are formed sequentially (Figure 1 of the manuscript), and it seems very unlikely that E_0 could lead directly to the fully mature form E_3 as it is the result of several hydrolytic events (i.e. the L3 loop and the N-terminal domain). When E_3 is formed, the transition of E_0 into E_3 (*via* E_1 and E_2) could be so fast that forms E_1 and E_2 are not experimentally observed. When inactive E_0 (C238A/H182A) was spiked with active E_3 , it is obvious that it is transformed to inactive E_3 -like *via* E_1 -like and E_2 -like intermediates (Supplementary Figure 12). Nevertheless, we cannot affirm that the maturation sequence is only and strictly E_0 to E_1 to E_2 and E_2 to E_3 . Indeed, intermediate maturation forms that we were not able to detect could exist in a transient manner i.e. $E_{1'}$ and $E_{1''}$ forms (Figure 5).

Figure 5. Description of the different maturation mechanisms of CgMCA-I. A. SDS-PAGE of CgMCA-I aliquots at maturation day 0 (D0), 4 (D4) or 8 (D8) in the presence or

absence of Ca^{2+} (10 mM) or EGTA (1 mM). The unmaturation form corresponds to the full-length protein and is referred to as the E_0 form, all different intermediate maturation forms are referred to as E_1 , $E_{1'}$, $E_{1''}$, E_2 and E_3 . **B.** Schematic representation of the maturation process of CgMCA-I (E_0 form) with the N-terminal part, the p20 domain, the linker and the p10 domain, respectively, colored in *blue*, *yellow*, *purple* and *green*. Two mechanisms of proteolytic cleavage can occur, the first resulting in the cleavage after residues Arg³⁰⁷, Lys²⁶³ and Arg⁵⁴ leading to the progressive appearance of E_1 , E_2 and E_3 forms of CgMCA-I, shown in the lower left part of the figure. The other mechanism involved two intermediate maturation forms after potential cleavage after residue Lys²⁹⁷ or Lys²⁹⁸ for the $E_{1'}$ form and after Arg²⁸¹ for the $E_{1''}$ form.

Question 7: Given the title of the manuscript (“maturation and activation”) I would expect more information about the role of the three cleavages after R54, K263, R307. The cleavage after R54, expected to happen last based on SDS-PAGE analysis, releases the Nterm peptide. It is stated that this peptide has no inhibitory role (line 307-309). While I agree that this conclusion is plausible by the structure, it could be validated by mutating the cleavage site. The cleavages at K263 and R307 release an inserting activation peptide, which is thought to be critical for activation.

We agree that the behavior of the Arg⁵⁴, Lys²⁶³ and Arg³⁰⁷ mutants could be indicative of the importance of these cleavages for the activation of the metacaspase. We had already tested the maturation process of mutants Lys²⁶³ and Arg⁵⁴, and in the course of the review, we also obtained and tested mutant Arg³⁰⁷ (we added a panel to Sup. Figure 4). We observed a lack of maturation inhibition for this new mutant. This could indicate that the cleavage is more structure dependent rather than being sequence dependent. Exposed loops could be easily targeted whatever the residues (basic or not) at the cleavage site. We are inclined to think that the presence of lysine or arginine is a guide for the autocatalytic activity of the enzyme, favoring cleavage at the right place for a proper, fast and efficient activation *in vivo*, but that the presence of a basic residue, at least in our case and at *in vitro* conditions, is not mandatory. In other words, it could be very difficult to block the maturation/activation of the enzyme at a certain point of the process. We added the result regarding mutant R307A to the manuscript, paragraph “**Analysis of amino-acid residues involved in CgMCA-I maturation**” and added a panel showing the R307A maturation in Sup. Figure 4 (in Supplementary Information).

In addition, we now have measured the enzymatic activity on the different mutants (Sup. Figure 14), and the results seems to confirm that the maturation indeed occurred but with certain specificities (Cf manuscript lines 311-332).

However, several questions remain:

a) Are both cleavages important to gain full activity?

When observing our model system, it seems that more than one cleavage could be necessary/mandatory to gain a full activity. It is *a priori* really difficult to separate the different intermediates of maturation as we were not able to stop the process of maturation. In this context, all intermediates (e.g. E₁ or E₂) cannot be tested independently to rule out the possibility that these partially matured forms could be fully active. However, our model of activation suggests that this hypothesis is not the most probable (see answer to question 6).

Alternatively, another possibility could be that a single cleavage, e.g. at position 54 which would remove the amino terminal part, could be sufficient to fully activate the enzyme. If so, this cleavage could occur for an infinitesimal part of the enzyme pool. Here again, it is difficult to prove in our case. However, in view of our results, this is not the most probable hypothesis.

Interestingly, it has been shown that processing of *Trypanosoma brucei* metacaspase 2 *TbMCA2* seems to modulate its substrate specificity. The uncleaved/non-processed form, the *TbMCA2* K55/268G mutant, shows activity towards small synthetic substrates but is unable to cleave larger substrates like azocasein⁷. For our system this may reflect that *CgMCA-I* shows a basal activity before maturation, namely the E_{low} forms (E₀, E₁ and E₂), prior to achieving a fully mature and active form, namely the E_{high} form (E₃).

To the manuscript we have added “Earlier studies reported different catalytic activities regarding the maturation level of metacaspase. For example, processing of *Trypanosoma brucei* metacaspase 2 *TbMCA2* seems to modulate its substrate specificity. The uncleaved/non processed form (*TbMCA2* K55/268G mutant) shows activity towards small synthetic substrates but is unable to cleave larger substrate like azocasein¹¹. This could illustrate the fact that *CgMCA-I* shows a basal activity before maturation, namely the E_{low} forms, prior to achieving a fully matured and active form, namely the E_{high} form.” to the **Discussion** of the manuscript

(lines 348-354, page 11).

This observation could suggest that metacaspases may have different cellular functions depending on their level of maturation/catalytic activity.

b) Does the sequence of cleavages matter, i.e. is the cleavage at R307 necessary for the second cleavage at K263 to occur?

In our manuscript, we show that the maturation seems to be sequential, or at least mostly sequential. So, a possibility would be that the successive cleavage sites could be more or less accessible depending on the maturation step and that a given cleavage could promote the access to the next cleavage site. In order to unequivocally answer this question, it would be necessary to block, at some point, the cleavage of at least one of the sites. Unfortunately, as stated above, the maturation process is probably not only sequence dependent, making the cleavages difficult to block.

However, the sequential nature of the maturation strongly suggests that the cleavage of a first site would not be mandatory for the cleavage of the next site but certainly would enhance or favor it.

c) Is cleavage a prerequisite for metal binding – or rather the other way, metal binding a prerequisite for cleavage? The sigmoid activity gain curve in Figure 6 could indicate a synergy of both effects.

When looking at the different publications which describe the processing of the metacaspase, it is shown that the cleavage is dependent on the presence of Ca^{2+} ^{1,8-10}. This is in agreement with our findings showing absence of maturation in the presence of EGTA (Figure 1 in the manuscript).

However, in order to verify if Ca^{2+} binds to metacaspase in the very first step of the maturation, we have performed nano Differential Scanning Fluorimetry (nano-DSF) assays to determine the affinity for Ca^{2+} at different incubation times and at different stages of maturation (see answers to question 5). These results have been added to the “**Enzymatic analysis of CgMCA-I maturation**” part of the manuscript (lines 311-320 pages 10-11). We also added the material

and methods paragraph concerning nano-DSF experiments in the **Material and Methods** section (“**Thermostability and calcium affinity**”), as well as to the **Supplementary Information document** (“Sup. Figure 13”).

Regarding the protein at maturation day 0, we observed a slight increase in the affinity at different incubation times (0, 1 hour and 24 hours) in the presence of calcium. At maturation day 8, the affinity for calcium does not change significantly with incubation time and is similar to that observed for the protein at day 8 indicating that the very first cleavages could slightly enhance the affinity for Ca^{2+} , which is already strong as soon as the protein is translated. Altogether, this indicates that the maturation is not a prerequisite for calcium binding but that calcium binding is mandatory for the enzyme maturation.

Question 8: Figure 3: I recommend to label the water molecule with H₂O rather than W.

We agree and now changed to H₂O.

Question 9: “As concerns CgL7 (including the α cap) it shows a most variable sequence, size or structural organization (Sup. Figure 3) should be sup Fig 2.

We are sorry but we are not sure to understand what the referee is suggesting. Our aim here was to demonstrate the lack of conservation for the CgL7 loop in the three determined MCA structures and thus referred to Sup. Figure 3 (Sup. Figure 6 in the revised manuscript: Primary- and secondary structure alignment of CgMCA-I^{Ca}, CgMCA-I^{Mg}, TbMCA-Ib et ScMCA-I). But maybe we have misunderstood. If so, could the referee be please more specific?

Question 10: Chemical hydrolysis. (for example, suppl fig 7). The reader would benefit it were be explicitly said that the chemical hydrolysis occurs is observed in the absence of a protease (which is what I assume).

As assumed correctly by the referee, chemical hydrolysis of the Z-GGR substrate occurs in the absence of any protease. This was previously mentioned in the manuscript by discussing the “first-order kinetic constant for the non-catalyzed (k_{chem})” (lines 268-269, page 9). To make this

clearer, we added in the following to **Supplementary Information**, first paragraph “**Enzyme kinetics and characterization**”: “The script takes into consideration the chemical hydrolysis of Z-GGR-AMC **in the absence of any protease**”.

Replies to reviewer 2:

In my opinion the article titled “Structural and molecular determinants of *Candida glabrata* metacaspase maturation and activation by calcium” by Conchou *et al.*, is very interesting with original data and very important for future yeast metacaspase studies. In my opinion the data are very conclusive and the text are good, I just make two little appointments that’s easy to be modified, and suggest that this manuscript be publish in **Communications Biology.**

We thank the reviewer for the kind words and hope that this work will be helpful for the future yeast metacaspase studies.

Question 1: In page 3, line 70 the authors says that “We identified for the first time a calcium binding site”, I disagree cause McLuskey *et al.*, they described a presence of this calcium binding site on *TbMCA-Ib* structure and the corresponding residues are conserved at *CgMCA-I*.

In fact McLuskey *et al.*,¹⁰ determined the first metacaspase structure in complex with a Samarium ion (Sm^{3+}). This observation incited the authors to assume that the Sm^{3+} binding site could have corresponded to the calcium binding site. However, as we demonstrated with both our experimental *CgMCA-I* in complex with Ca^{2+} ion or Mg^{2+} ions, Sm^{3+} is more likely to mimic the Mg^{2+} binding site in *CgMCA-I*. However, we agree that the binding site observed for the Sm^{3+} ion is in vicinity of that observed for the Ca^{2+} ion. In order to make this clearer, we have changed our sentence to “We observed for the first-time electron density corresponding to a calcium ion, thereby allowing the identification of a calcium binding site, in the vicinity of a previously observed samarium ion binding site¹⁴.” (lines 70-72, page 3).

Question 2: The legend of Figure 4 must be rewritten, because it is necessary to spend a lot of time to understand which structure is from *CgMCA-I*^{Ca} and which is from *CgMCA-*

I^{Mg}. I suggest that the figure be redone by differentiating the 2 enzymes in a single color to identify them.

We agree that the colors were confusing. As suggested, we choose a single color for each structure and changed the legend accordingly (Figure 4 in the manuscript).

Replies to reviewer 3:

The authors Conchou *et al.*, describe maturation of metacaspase CgMCA-I from *Candida glabrata* and the role Ca²⁺ ions play in it. Unfortunately, I did not enjoy reading this manuscript. On the one hand it is loaded with technical details and data, on the other hand it lacks description and explanation of certain technical details and data, which raise concerns about technical soundness of the work and correctness of conclusions. In addition, this manuscript lacks perspective of biological importance of the work and a related conclusion. I hope that my concerns are only the problem of presentation and which the authors will be able to address them in the revised version.

We are sorry that the reviewer did not find the manuscript clear enough and insufficiently documented regarding the conclusions of this work and the potential biological perspectives. In order to make the reading of the manuscript easier and to strengthen the conclusions, we have given precisions to certain protocols, added several new experiments (see answers to reviewer #3 and other reviewers) and went deeper into the conclusion in terms of biological aspects. We sincerely hope that this amended version will be more suitable.

Question 1: The Ca and Mg crystal structures of CgMCA-I structures were determined at various resolutions. Table 1 provides cell constants at various precision, with cell length of a space group C2 without a decimal point.

We apologize for this error. The discrepancies in the description of the parameters in Table 1 have now been corrected and homogenized.

The Ca bound form contains 4 ligand atoms and 7 solvent molecules which all have lower average B-factors than protein atoms. yet it does not explain what they were. Are there four Ca²⁺ ions?

We are sorry for this lack of clarity. Two Ca^{2+} ions are present, one per molecule in the asymmetric unit. The two other molecules are chloride ions coming from the crystallization conditions. The corresponding electron density maps of the first coordination sphere ($2\text{Fo}-\text{Fc}$ at 1.25 sigma), as well as the omit-map for the Ca^{2+} ions are shown below in Figure 6, 7, 8, 9. In addition, Ca^{2+} ions were the only divalent cations in the conditions that was used for crystallization and the geometry and the coordination bond lengths are consistent with a calcium ion.

Figure 6. First coordination sphere of Ca^{2+} in subunit A of CgMCA-ICa .

Figure 7. Omit-map (5 sigma) subunit A of CgMCA-I^{Ca}.

Figure 8. First coordination sphere of Ca²⁺ in subunit B of CgMCA-I^{Ca}.

Figure 9. Omit-map (5 sigma) subunit B of CgMCA-I^{Ca}.

The Mg from diffracted to 1.6Å resolution, it has 36 ligand atoms or ions, yet their average B-factor is 50.2, which is twice as much as average protein B-factor and higher than average B-factor of solvents. Are ligands and ions indeed what the authors claim to be, is their occupancy lower than 1.0? The later is a red flag which raises concern about the correctness as interpretation of ions constitutes some of the core arguments of the conclusions. The PDB validation report, though included, is suggestive, but the authors should address this issues in the manuscript. Maybe even to re-refine the structure. Comparison of B-factors of ions with their surroundings in a crystal diffracting to 1.6Å should provide an unambiguous answer about the number of electrons of each ion. Please elaborate and show their electron density maps.

Sorry for that. Actually, the B factor of 50.2 corresponds to the 6 magnesium ions and two Hepes molecules observed in the structure. We have recalculated the B-factor for each Mg²⁺ and it appears that they are consistent with the B factors of their surrounding atoms (water molecules or residues coordinating the ion). This is summarized in Table 5 below. The problem comes from the Hepes molecules for which the B factor (53.45 for the Hepes in monomer A and 54.01 for Hepes in monomer B) is indeed twice the value of the average protein B factor (25.6 for the protein), which causes a bias in the reported B factor for ion/ligands (32.26 for magnesium ion / 53.73 for Hepes molecules). Moreover, the residues and water molecules surrounding a given magnesium present similar B factor values as the ion (Table 5).

Mg3 (chain B ; same as Mg6 in chain A) , Mg B factor: 28.3	Mg6 (chain A; same as Mg3 in chain B), Mg B factor: 35.28
H ₂ O 599 2.12 27.52	H ₂ O 426 2.16 44.63
H ₂ O 226 2.16 25.93	H ₂ O 508 1.94 34.33
H ₂ O 224 1.29 27.43	H ₂ O 492 2.22 41.32
H ₂ O 229 2.01 30.23	H ₂ O 230 2.11 31.12
H ₂ O 240 2.26 35.44	H ₂ O 160 2.24 33.37
ASP214 OD2 2.14 24.34	ASP214 OD2 2.15 27.23
Waters and residues Average B factor: 28.48	Waters and residues Average B factor: 34.67
Mg2 (chain B; same as Mg5 in chain A), Mg B factor: 25.6	Mg5 (chain A; same as Mg2 in chain B), Mg B factor: 33.69
H ₂ O 235 2.17 28.78	H ₂ O 431 2.44 36.31
H ₂ O 221 2.07 21.43	H ₂ O 393 2.26 36.8
H ₂ O 232 2.08 28.8	H ₂ O 178 2.06 36.55
H ₂ O 222 2.13 19.94	H ₂ O 26 2.17 23.07
H ₂ O 231 2.10 28.11	H ₂ O 134 2.21 27.25
H ₂ O 233 2.27 31.49	H ₂ O 395 2.16 33.37
Waters and residues Average B factor: 22.05	Waters and residues Average B factor: 32.24
Mg1 (chain A; same as Mg4 in chain B), Mg B factor: 41.87	Mg4 (chain B; same as Mg1 in chain A), Mg B factor: 28.84
H ₂ O 581 1.94 46.77	H ₂ O 377 2.07 34.66
H ₂ O 391 2.33 40.46	H ₂ O 365 2.08 29.42
H ₂ O 392 2.07 32.43	H ₂ O 367 2.14 37.48
H ₂ O 98 2.32 35.8	H ₂ O 366 2.04 26.63
ASP 190 OD1 2.08 45.33	ASP190 OD1 2.18 33.46
ASP 188 OD1 2.06 37.46	ASP188 OD1 2.08 29.44
Waters and residues Average B factor: 40.37	Waters and residues Average B factor: 31.85

Table 5. B factors. For each magnesium, the water molecules and the residues participating in the first coordination sphere are shown. The second column corresponds to the distance (Å) and the third column to the B factor (Å²).

In addition, we show images of electron density maps of the first coordination sphere (2Fo-Fc at 1.25 sigma) for each ligand (Figures 10-25 in this rebuttal), as well as the omit-map for the Mg²⁺ ions. Geometry and coordination bond lengths are consistent with magnesium ions.

Furthermore, the concentration of magnesium in the crystallization condition is 200 mM, strongly suggesting that Mg²⁺ is a *bona fide* option.

Figure 10. First coordination sphere of Mg^{2+}_A in subunit A of CgMCA-IMg .

Figure 11. Mg^{2+}_A Omit-map (5 sigma) subunit A of CgMCA-IMg .

Figure 12. First coordination sphere of Mg²⁺_B in subunit A of CgMCA-I^{Mg}.

Figure 13. Mg²⁺_B Omit-map (5 sigma) subunit A of CgMCA-I^{Mg}.

Figure 14. First coordination sphere of $\text{Mg}^{2+}_{\text{C}}$ in subunit A of $\text{CgMCA-I}^{\text{Mg}}$.

Figure 15. $\text{Mg}^{2+}_{\text{C}}$ Omit-map (5 sigma) subunit A of $\text{CgMCA-I}^{\text{Mg}}$.

Figure 16. First coordination sphere of Mg^{2+}_A in subunit B of $\text{CgMCA-I}^{\text{Mg}}$.

Figure 17. Mg^{2+}_A Omit-map (5 sigma) subunit B of $\text{CgMCA-I}^{\text{Mg}}$.

Figure 18. First coordination sphere of $\text{Mg}^{2+}_{\text{B}}$ in subunit B of $\text{CgMCA-I}^{\text{Mg}}$.

Figure 19. $\text{Mg}^{2+}_{\text{B}}$ Omit-map (5 sigma) subunit B of $\text{CgMCA-I}^{\text{Mg}}$.

Figure 20. First coordination sphere of $\text{Mg}^{2+}_{\text{C}}$ in subunit B of $\text{CgMCA-I}^{\text{Mg}}$.

Figure 21. $\text{Mg}^{2+}_{\text{C}}$ Omit-map (5 sigma) subunit B of $\text{CgMCA-I}^{\text{Mg}}$.

Figure 22. Density map of HEPES at 1.25 sigma in subunit A of *CgMCA-IMg*.

Figure 23. HEPES Omit-map (5 sigma) subunit A of *CgMCA-IMg*.

Figure 24. Density map at 1.25 sigma in subunit B of *CgMCA-IMg*.

Figure 25. HEPES Omit-map (5 sigma) subunit B of *CgMCA-I^{Mg}*.

Question 2: The authors describe details about fitting of parts of the model to electron density maps, yet they show no maps in the figures 2C and B, E, F, and A (lines 136, 151, 170).

We agree that density maps for the most important parts of the structure, in particular the one we used to sustain our conclusion, should be added. Electronic densities 2Fo-Fc and Omit-maps have been and added to the Supplementary Information (**Sup. Figure 8 and Sup. Figure 9**). Density maps for loops and extremities of missing parts of the protein are now included with the following legends:

Supplementary Figure 8. Electronic density showing the extremity of missing loops as seen in *CgMCA-I^{Ca}* and in *CgMCA-I^{Mg}*. **A.** Missing amino-terminal part Met¹-Gly⁶⁸ in *CgMCA-I^{Ca}*. **B.** Missing loop Asn²⁶⁰-Ile³⁰⁹ in *CgMCA-I^{Ca}* **C.** Missing amino-terminal part Met¹-Ser⁷² in *CgMCA-I^{Mg}*. **D.** Missing loop Asn²⁶⁰ to Ile³⁰⁹ in *CgMCA-I^{Mg}* **E-F.** Absence of loop Gly⁸⁴-Gln⁸⁷ in *CgMCA-I^{Ca}* (**E**) as opposed to *CgMCA-I^{Mg}* (**F**).

Supplementary Figure 9. First coordination sphere of Ca²⁺ in subunit A of *CgMCA-I^{Ca}* (A) or subunit B of *CgMCA-I^{Ca}* (B). Omit-map (5 sigma) around Ca²⁺ in subunit A (A) and B (B) of *CgMCA-I^{Ca}*.

Question 3: The use of activation and maturation mechanisms is confusing (line 157). I suggest that the authors choose either maturation or activation, not both, otherwise they have to explain the difference between the two mechanisms. I prefer activation.

Our initial questioning, was does the maturation process of the protein (cleavage of the C-terminal, internal L6 loop and N-terminal domain) correlate with the activation of *CgMCA-I* (increase of the catalytic activity towards *Z-GGR-AMC* substrate). Both phenomena correlate/are linked to each other and are therefore difficult to separate. Consistent with previous publications about metacaspase (*ScMCA-I*¹ and *TbMCA-Ib*¹⁰) the authors also distinguish the activation phenomenon and the maturation phenomenon to which they referred by using the term auto-processing.

In order to be less confusing, we propose to explain at the very beginning of the manuscript the difference between the two terms and so, the following sentence has been added (lines 74-79, page 3): “Throughout, the term maturation is used to describe the different proteolytic cleavages that the metacaspase will undergo and, by extension, the different intermediates of maturation termed E_0 to E_3 . The term activation corresponds to the increased catalytic activity of the metacaspase which harbors either low activity during the first step of maturation “ E_{low} ”, or high activity at the last step of maturation “ E_{high} ”.”

Question 4: Then there is the issue of the number of steps in the activation mechanism. The authors observed 3 cleavages and a Ca^{2+} ion dependent step, which makes 4 steps and not 2.

According to what has been said in the discussion part of the manuscript, all of our results allowed us to provide functional and structural evidence on the Ca^{2+} dependency of metacaspases for their maturation and activation. A complete maturation demands at least three cleavages occurring at Arg³⁰⁷ (at one end of the L6 loop), Lys²⁶³ (departure of L6 loop) and Arg⁵⁴ (departure of N-terminal domain). We show that these three cleavages are Ca^{2+} dependent and are accompanied by a structural reorganization of the active site of *CgMCA-I* (Figure 4). Actually, the Ca^{2+} can be bound to the enzyme all along the process and the activation is not an additional step (see our results for *CgMCA-I* affinity for calcium, #reviewer 1).

According to our model, the enzyme is activated when it passes from its E_{low} forms (E_0 , E_1 and E_2 intermediates of maturation) to its E_{high} form (E_3 intermediate of maturation).

Altogether, the maturation occurs in three steps corresponding to two main activation states.

We are sorry for this confusion which may come from the above-mentioned maturation and activation terms which can be actually confusing. We hope that by explaining the two terms at the beginning of the publication, it will no longer be ambiguous.

It is possible that there are additional steps associated with separation of N- or C-terminal domains following the cleavage, hence potentially there is room for a quite complicated mechanism likely taking place in the first 2 to 4 days unobserved in Figure 6?

We agree with the referee that a more complicated phenomenon of maturation could exist. We could not exclude that the maturation sequence is strictly E_0 to E_1 , E_1 to E_2 and E_2 to E_3 . Indeed, intermediate maturation forms could exist in a transient manner i.e. E_1' and E_1'' forms (Figure 5, #reviewer 1).

However, we would like to add a clarification regarding the separation of the C-terminal domain as suggested by the reviewer. Actually, the C-terminal corresponding to p10 will not be liberated from the rest of the protein as it is composed of the two beta strands 7 and 8 which are embedded in the central beta sheet preventing any separation in solution (at non-denaturing conditions of course). We suppose that the reviewer was making references to the central part of the protein, fragment K263-R307.

What is the biological relevance of an 8 days long activation process of an enzyme? Such long periods are usually associated with transformation of the structure such as fibril formation and not with conformational change of a loop? Where are the controls of stability of components after 8 days of measurement? A similar publication on MCA-IV revealed Ca dependent activation mechanism which lasted minutes (Zhu et al., 2020, Nat. Comm.) Hence, a much better explanation of activation measurements and activation mechanism than provided is required to satisfy my concerns.

We agree that it seems unlikely that a biological mechanism (i.e the autoprocessing of CgMCA-I) that is part of a cell death phenomenon could take place in eight days. The length of this process could be explained by the fact that these experiments were performed *in vitro* and not *in vivo*. It cannot be excluded that in the cell, protein partners, metacaspase or Ca²⁺ local concentrations, specific cellular localization ... could accelerate the process.

It is also true that the conditions of maturation are different considering the available studies. Regarding purified ScMCA-I¹, it seems that the time of incubation is overnight. Regarding purified AtMCA-2d⁸, the time of incubation is extremely short (10 minutes), but only a very small fraction of the protein is matured after this time (Figure 4A of the publication). When *Arabidopsis thaliana* metacaspases are expressed in yeast (AtMCP1b or AtMCP2b) the protein's maturation in the cells seems to take around 72 hours⁹ (Figure 1B of the publication). On the other hand, regarding the time of maturation of *Trypanosoma brucei* TbMCA-Ib, it seems to be shorter or even significantly shorter. In the publication of McLuskey *et al.*,¹⁰ a complete maturation could take 16 hours to observe a complete disappearance of the “non-processed form” (Figure 6 of the publication). In the publication by Moss *et al.*,¹¹ the incubation is extremely short, only 30 minutes (Figure 2C of the publication).

Altogether, it shows that the maturation time is highly dependent on the surroundings. However, we would also like to emphasize that this particularly slow maturation process allowed us to decipher the sequential nature of this maturation. It also indicates that an *in vivo* study in the future would hopefully give us information on the cellular process at stake.

Question 5: The supplement contains no references. I believe that this is a glitch of the publisher's processing system as we had the same problem a while ago.

We are sorry for this oversight, we added the references of the Supplementary Information at the end of the document.

Replies to reviewer 4:

The authors report the first study of a type I metacaspase in the active form with bound calcium ions. Surprisingly, structures were obtained also with magnesium ions in it. Below

I summarized my comments. I think this is an interesting structure, but the manuscript itself needs some rewriting and reformatting of the figures.

Question 1: Cloning and purification of the WT protease

What construct was used for mutagenesis, CgMCA-I(His) or (His)CrMCA-I? Additionally, it is not clear what the expected size of MCA is. From the affinity chromatography elutions it is not clear which peak corresponds to the full-length size protein. Please, mark it with an arrow.

We are sorry for this oversight, we forgot to mention that point mutagenesis experiments were done using the construct of CgMCA-I with a His-tag at the C-terminal (CgMCA-I^{His}). To clarify the expected size of the different construction, we suggest adding a Table to the Supplementary Information (**Supplementary Table 2**) indicating the size of the recombinant protein (with His-tag) for both constructs.

Regarding the affinity chromatography elution, we now indicate the full-length protein with an arrow. Furthermore, for the size exclusion chromatography, considering that the maturation of CgMCA-I occurs during the purification steps (affinity and gel filtration) and knowing that these two steps are separated by a concentration step that may enhance the maturation process, the full-length protein peak is no longer detectable on the gel filtration chromatogram. According to our column S200 Increase 10/300, the retention volume for peak A is 17.04 mL corresponding to an apparent molecular mass of 27 kDa. Most likely, it mainly corresponds to the partially matured E₂ form of CgMCA-I which still possesses its amino-terminal part which contains the His-tag. This is confirmed by the analysis of fraction 6 (peak A) by Western-blot which has been added to **Supplementary Figure 2**. Regarding the analysis of fraction 10 by Western-blot, it appears that peak B corresponds as well to the amino terminal part (containing His-tag). That means that the last cleavage has already occurred at least for a fraction of the metacaspase population and that the form E₃ is already present in very small amounts.

Interestingly, both prominent bands disappear from the SDS-PAGE gel after SEC. It is very hard to understand how the protein, eluted from the SEC could be used for maturation studies (Fig. 1), since it is not in full-length anymore. Was protein differently prepared and eluted for this experiment? If yes, this should be made clearer in the main text.

As mentioned above, disappearance of both prominent bands, FL CgMCA-I (48 kDa) and the E₁ form of CgMCA-I (cleavage after Arg³⁰⁷ (38 kDa)) can be explained by the duration of affinity- and size exclusion chromatographies. In addition, chromatographies are separated by a concentration step prior to sample loading onto the gel filtration column. Considering the time needed to perform affinity chromatography (almost 2h), concentration (1h) and elution on gel filtration (almost 1h), the continuing auto-processing of CgMCA-I results in the disappearance of the first bands on the SDS-PAGE.

In order to overcome the time constraints for the study of the enzyme maturation phenomenon, we did not use the same enzyme purification protocol and technique. We performed Biosprint purification, a technique using Ni-NTA-coated magnetic agarose beads and that can be done in 40 minutes as described in **Supplementary Information**. The use of magnetic beads allows us to perform a rapid and efficient one step purification. After purification, we collected aliquots at day 0 and thus visualized the full length CgMCA-I.

In order to make this clearer to the reader, we added “In order to gain a better understanding of CgMCA-I maturation, we purified by affinity chromatography using Biosprint 96 (see **Maturation Assay, SI**) and analysed by gel electrophoresis the wild type protein after 0, 4 and 8 days, respectively, in the presence or the absence of calcium or EGTA to observe the maturation process over time and the impact of Ca²⁺ ions on this process (Figure 1).” in the manuscript (lines 88-92, pages 3-4).

Question 2: I understand that authors wanted to show the structures as they are in the asymmetric unit, but it makes no sense for a common reader. Rather move those to the supplement and in Fig. 2 show monomeric units of CgMCA-I^{Ca} and CgMCA-I^{Mg} in the same orientation for the comparison.

We agree that a representation with the two monomers in the asymmetric unit makes it more difficult to compare. We changed Figure 2 (now Figure 3) and showed only one monomer for both structures. The representation with two monomers in the asymmetric unit is now in Supplementary Figures (Sup. Figure 5).

Also in the text, it would be good to start with describing what is similar and then proceeding to what is different.

We start now the “**Overall structural features of CgMCA-I**” section by a paragraph explaining the similarities between the two structures:

“CgMCA-I^{Ca} crystallized in the trigonal space group *P*32 with two molecules in the asymmetric unit (Table 1, Sup. Figure 5 A) [...] Cys²³⁸ and His¹⁸² located on loops CgL4 and CgL3, respectively, from the catalytic dyad (Figure 3 A, B, C, F).” (line 164-174, page 6 in the manuscript).

Then, we continue with the specificities of each structures with the following paragraph:

“Regarding the specificities of CgMCA-I^{Ca} [...]” (lines 175-202, pages 6-7 in the manuscript).

Question 3: General presentation of results

The paper starts with metals, continues with structures and goes back to maturation. I think the manuscript requires rewriting, to start with the maturation (so that the reader understands well what is happening to the protease while it “matures”-here it is crucial to define what this term means.

Our initial questioning, i.e does the maturation process of the protein (cleavage of the C-terminal, internal L6 loop and N-terminal domain) correlate with the activation of CgMCA-I (increase of the catalytic activity towards Z-GGR-AMC substrate). Both phenomena correlate/are linked to each other and are therefore difficult to separate. Consistent with previous publications about metacaspase (*Sc*MCA-I¹ and *Tb*MCA-Ib¹⁰), the authors also distinguish the activation phenomenon and the maturation phenomenon to which they referred by using the term auto-processing.

In order to be less confusing, as also written above, we now explain at the very beginning of the manuscript the difference between the terms maturation and activation. Hence, the following sentence has been added (lines 74-78, page 3): “Throughout, the term maturation is used to describe the different proteolytic cleavages that the metacaspase will undergo and, by extension, the different intermediates of maturation termed E₀ to E₃. The term activation corresponds to the increased catalytic activity of the metacaspase which harbors either low

activity during the first step of maturation “E_{low}”, or high activity at the last step of maturation “E_{high}”.”

Regarding the organization of the manuscript, and as suggested by the reviewer, we first describe in more detail the maturation process by putting the paragraph “**Analysis of amino-acid residues involved in CgMCA-I maturation**” just after the paragraph “**Cation dependent maturation of CgMCA-I**”. We have also introduced in the paragraph “**Analysis of amino-acid residues involved in CgMCA-I maturation**” the mass spectrometry analysis. By doing this, we describe the whole maturation process in a row.

Regarding the enzymatic experiments, it is difficult to put it earlier as these results interpretation needs the structures to be understandable. We hope that this reorganization nonetheless is satisfactory for the reviewer.

Question 4: Effect of calcium

In the case of R54A mutation, in line 230 the authors write that the process is stopped before the last step. But from the Fig. 1 I guess one could speculate what does not happen. This should be elaborated. In the case of R263A mutation, yes, the protein is degraded, but in a different manner than the WT. This should at least be discussed.

We agree with the reviewer that the description of the R54A mutant autoprocessing can be confusing. Actually, the last two cleavages do not seem to appear as the band corresponding to the p20 (corresponding to E₃ form) is not visible for the R54A mutant neither is the M1-K263 fragment (corresponding to E₂ form) (Figure 1 B in the manuscript). What might be confusing is that, comparing to the wild type, two new intermediate forms of maturation, namely E₁' and E₁"', seems to appear. These two forms could exist for the wild type but are less visible. As mentioned by #reviewer 3, there is room for a more complicated maturation process (Figure 5, #reviewer 1) and more intermediates could exist that we did not have considered in the description of the maturation mechanism of CgMCA-I. Thus, in this case, we see that the R54A mutant could enhance the formation of new intermediate forms of maturation of CgMCA-I. Altogether, beyond the fact that the R54A mutant does not undergo the last maturation step compared to the wild-type (or that the last cleavage at position 54 is delayed), we think that the difference observed on the gels would rather come from the fact that these transient forms of

maturation ($E_{1'}$ and $E_{1''}$) are rapidly cleaved to lead to the E_2 form with the WT enzyme (Figure 26).

Figure 26. Comparison of the maturation of WT *CgMCA-I* and *CgMCA-I* R54A mutants. SDS-PAGE of WT (left) and R54A mutant (right) at days 0 (D0), 4 (D4) and 8 (D8) of maturation in presence or absence of Ca^{2+} (10 mM) or EGTA (1 mM).

Altogether, in our opinion, it reinforces the hypothesis that the mutation of residue Arg⁵⁴ leads to the interruption/slowing of the maturation process. Therefore, we agree that it could be modulated in the manuscript. So, we change “It appears that even in the presence of calcium the process seems to be stopped before the last step of maturation”, by “It appears that in the presence of calcium the process is delayed, leading only to intermediate forms of maturation without reaching a fully mature form of *CgMCA-I* after eight days, confirming that cleavage at position R54 is most likely the last one to occur” (line 131-134, page 5).

The results obtained by K263D and K263F mutants are very intriguing. What would be needed here are activity measurements using fluorogenic substrates.

After purification, all mutants were tested for activity at day 0, 4 and 8 following the same protocol as for the wild type (the same amount of protein checked by Nanodrop was added for these assays).

Regarding the K263 mutants, we systematically observed a delayed activation. K263F and K263W never reached the activity maximum observed for the wild type, suggesting an aberrant maturation at some point. Surprisingly, the K263A mutant, despite the maturation delay, shows a higher activity at day 8. The reason for which this mutant is more active after a complete maturation is however not clear so far. Here again, an explanation could be that the altered cleavage pattern, due to the mutation, may induce a more active subpopulation of enzyme. Mutant R307A harbors a quite unexpected maturation process with a sudden activation at day 4 and a decrease at day 8 maybe indicating a degradation of the protein by itself due to an incorrect maturation.

Finally, mutant R54A revealed an unexpected high activity all along the maturation process (2 times the wild type activity).

For metacaspase 2 from *Trypanosoma brucei* (TbMCA2) it has been suggested that the N-terminal plays a role in the control of the activity by inhibiting the enzyme. Surprisingly, our current studies of CgMCA-I indicate that the Nter is the last fragment to be cleaved during the maturation process, and which may, a priori, rule out the possibility of an inhibitory control by this latter.

Nevertheless, a hypothesis could be that the role of the N-terminal of CgMCA1 is not to inhibit the enzyme but rather to regulate its activity. Indeed, in the very last step of maturation which corresponds to the departure of the Nter fragment (E₂ to E₃), we observed a rapid increase in the activity (E_{high}). Hence, mutation of R54A could alter the role of the Nter by *e.g.* modifying or impairing interactions with the rest of the protein?

Altogether, it appears that the maturation process of the metacaspase is not an “on-off” phenomenon, but rather a subtler process “tuned” for getting the fully active enzyme at the right time. These results also confirmed that the presence of basic residues at the cleavage sites is not mandatory, at least *in vitro*, and that the cleavage is probably more structure dependent.

We added these results to the Supplemental Information (Sup. Figure 14), and added the following sentence (lines 321-332, page 11) to the manuscript: “The cleavage site mutants all display different behavior from a catalytic point of view. Mutants K263A, K263F, K263W and R307A all showed a delay in the activation process. Furthermore, K263F and K263W are less

active after eight days of maturation. For mutant R307A, a rapid activation at day 4 but a decreased activity at day 8 was observed, illustrating an aberrant maturation. Mutant R54A showed the most intriguing behavior with a higher activity than that of the wild type enzyme throughout the maturation process, and could suggest that the N-terminal, when un-mutated, could modulate the metacaspase activity prior to cleavage in the very last step of the maturation. These results also confirmed that the presence of basic residues at the different cleavage sites, at least *in vitro*, are not mandatory to mature the protein, and that the cleavage is probably more structure dependent. However, the anarchic levels of activity obtained for the different mutants suggest that a properly matured metacaspase requires the presence of these residues.”

Question 5: Activity

E3 can not be the active form. It is well known that p10 domain is needed for the activity, that is why it is also present in the structure. Please re-label the Fig. 1 and re-write the enzymatic maturation claims. Cleavage does not release the p10 domain, but most probably rearranges it.

We apologize for the lack of clarity in our explanation. It is true that the only p20 domain that we label E₃ on the figure cannot be the active form without the p10 domain. We keep the notation E₀, E₁, E₂ and E₃ all along the manuscript but removed those terms in Figure 1 B. To make it clearer for the reader we added a panel on Figure 1 with the structural representation of these different forms E₀, E₁, E₂ and E₃ corresponding to what can be observed on the SDS-PAGE under denaturing conditions.

Question 6: The 280 loop

It should be discussed, that in *TbMCA-Ib*, the 280-loop moves towards and not away from the active site in the presence of calcium, so it can not be active as a capping loop as suggested here (lines 324-326).

We agree that the mechanism we are confronted to is very complex and it is also true that we must be very cautious in order to avoid any overstatement. First, it is important to note that the two loops are different in terms of sequence and size. However, despite their differences, we can say that they are both very flexible (none of them is fully visible in density in the structure), this flexibility probably contributing to their function during the catalysis.

In *TbMCA-Ib*, the movement of the loop toward the active site in presence of calcium is deduced from the movement of only 4 residues¹⁰, the rest of the loop being not visible, so we can imagine many possibilities.

In *CgMCA-I*, the presence of calcium seems to take away the loop from the active site, but here again, the exact position of the loop in presence of calcium is not sure, it could stay in the vicinity of the active site but without stabilization, interactions between residues Thr³³¹-His¹⁸² and Asn³³⁷-Ser³⁷³ disappearing (Sup. Figure 15 in Supplementary Information).

Question 7: Cleavages

I do not understand where does the first cleavage occur, since in line 299 the authors claim that the p10 domain “is released”; but there is no cleavage marked and the cleavages denoted as the ones between the p20 and p10 domains are “the second and the third” cleavage. Fig. 1 should be merged with Fig. S6 as it is very important. Additionally, I would suggest making the R307 mutant; again, complementing all the mutant characterisation with the activity assays.

We are sorry for this overstatement, actually, the first cleavage occurs at position Arg³⁰⁷ followed by the cleavage at position Lys²⁶³. The C-terminal fragment corresponding to the p10 will not be freed from the rest of the protein as it is composed of the two beta strands 7 and 8 which are embedded in the central beta sheet preventing any separation in solution (at non-denaturing condition of course (amended Fig. 1)). p10 can only be released from the rest of the protein at denaturing conditions as soon as the cleavage at position Arg³⁰⁷ occurs and this was what we referred to, but it was not properly explained. So, we changed the sentence to “forming an independent domain that can be released from the rest of the protein in denaturing condition” (lines 337-339, page 11 in the manuscript). On the other hand, the fragment K263-R307 is indeed released from the rest of the structure even in solution. To note, we performed a structural prediction of the protein with Alpha fold 2 and interestingly, this fragment K263-R307, even if structured in two main alpha helices, does not seem to interact with the rest of the protein due to its particular folding (Figure 27). This could indicate that in solution, fragment K263-R307 could be separated from the protein very fast.

Figure 27. Superposition of CgMCA-I^{Ca}, CgMCA-I^{Mg} structures with the CgMCA-I AlphaFold 2 model. CgMCA-I^{Ca}, CgMCA-I^{Mg} and AlphaFold2 models are respectively colored in *green, orange* and *blue*.

Question 8: Putting this work in the wider context:

- activation in trans should be discussed with similar observations made by Watanabe and Lam (2005)

We now added the following paragraph regarding *trans* cleavage observed by Watanabe and Lam (lines 368-375, page 12 in the manuscript):

“As described previously, the cleavage of the metacaspase could exist in *trans* as observed by Watanabe and Lam for rAtMCP2d⁸. However, the authors suggested that cleavage in *cis* could exist as well in an even more efficient way. Interestingly, this observation is in agreement with our most probable model. Indeed, it could illustrate the fact that passage from E₀ to E₁, E₁ to E₂ and E₂ to E₃ would be in *cis*. Then, as soon as an E₃ form (E_{high} form) appears, the *trans* cleavage would be the most probable. In the cell, one could imagine that the autocleavage could generate a E_{high} form able to transactivate the resting pool of metacaspase.”.

- the importance of the p280 loop (L7 loop) as a calcium binding site by Klemenčič and Funk (2018)

In the discussion, we had briefly mentioned the p280 loop and cited Klemenčič and Funk (2018): “Finally, in the study of *TbMCA-Ib*, the authors suggested that a second site of low

affinity^{3,20,21} could exist for Ca²⁺.” However, we have reinforced the discussion by adding the following sentences (lines 401-405, page 13 in the manuscript): “Finally, in the study of *TbMCA-Ib*, the authors suggested that a second site of low affinity^{3,20,21} could exist for Ca²⁺ playing a role in the control of the catalytic mechanism. In addition, the authors suggest that the binding could be dependent on the negatively charged residues in the p10 domain. In our structure, we find that the Mg²⁺ ion is embedded in a negatively charged pocket formed by residues rather coming from the p20 domain like Asp¹²⁰ (Sup. Figure 16).”

- the unobserved cleavage between the p20 and p10 domains *in vivo* by Coll *et al.*, 2010

We have now added a short paragraph in the discussion section and added the reference by Coll *et al.*, 2010. We discuss the role of the metacaspase cleavage and maturation in an *in vivo* context (lines 349-358, page 12):

“Earlier studies reported different catalytic activities regarding the maturation level of metacaspase. For example, processing of *Trypanosoma brucei* metacaspase 2 *TbMCA2* seems to modulate its substrate specificity. The uncleaved/non processed form (*TbMCA2* K55/268G mutant) shows activity towards small synthetic substrates but is unable to cleave larger substrate like azocasein¹¹. This could illustrate the fact that *CgMCA-I* shows a basal activity before maturation, namely the E_{low} forms, prior to achieving a fully matured and active form, namely the E_{high} form. More surprisingly, Coll *et al.*,¹⁸ showed that, in *Arabidopsis thaliana*, the function of *AtMC2* which is to negatively regulate cell death, does not depend on its catalytic residues¹⁸. This indicates that the cellular function of metacaspase could go beyond a simple maturation and catalytic activation.”

Question 9: Minor

Line 87: what is meant by complete maturation?

Sorry for the lack of explanation or clarity, actually, when we talk of complete maturation, we are referring to the E₃ form, namely, the p20 and p10 after all cleavages occurred. We changed the sentence to “In the absence of calcium or EGTA, *CgMCA-I* undergoes cleavage and starts to mature at day 0, continuing until day 8 without reaching a complete mature form namely the E₃ form (*vide infra*).” (lines 93-95, page 4 in the manuscript).

p20 and p10 are sometimes written as P20 and P10, please unify.

We have now change to p20 and p10 all along the text.

Line 136: change Asn360 to Asn260.

It has been now changed, sorry for the error.

Line 162: please write which MCA (AtMCA-IIa).

It has now been corrected.

Line 228: departure of the N-terminal?

We have change departure of the N-terminal by “On the other hand, the presence of a band at 48 kDa in the western blot until day 8 suggests that the cleavage of the N-terminal part of the protein occurs at the very end of the process (Figure 2 A, B, C).” (lines 128-140, page 5 in the manuscript)

Line 255: this is not a caspase.

It has been corrected and changed to “At day 0, the k_{cat} value is $2.8 \pm 0.7 \text{ min}^{-1}$ and reaches $15.8 \pm 0.6 \text{ min}^{-1}$ at day 8 indicating that the matured metacaspase is 5 to 6 time more active than non-matured form” (lines 271-273, page 9).

References

1. Wong, A. H.-H., Yan, C. & Shi, Y. Crystal Structure of the Yeast Metacaspase Yca1 *♦. *Journal of Biological Chemistry* **287**, 29251–29259 (2012).
2. Reichle, R. A., McCurdy, K. G. & Hepler, L. G. Zinc Hydroxide: Solubility Product and Hydroxy-complex Stability Constants from 12.5–75 °C. *Can. J. Chem.* **53**, 3841–3845 (1975).
3. Franke, D. & Svergun, D. I. DAMMIF, a program for rapid ab-initio shape determination in small-angle scattering. *J Appl Cryst* **42**, 342–346 (2009).
4. Konarev, P. V., Volkov, V. V., Sokolova, A. V., Koch, M. H. J. & Svergun, D. I. PRIMUS: a Windows PC-based system for small-angle scattering data analysis. *Journal of Applied Crystallography* **36**, 1277–1282 (2003).
5. Carraro, M. & Bernardi, P. Calcium and reactive oxygen species in regulation of the mitochondrial permeability transition and of programmed cell death in yeast. *Cell Calcium* **60**, 102–107 (2016).
6. Machado, M. F. M. *et al.* Substrate specificity and the effect of calcium on Trypanosoma brucei metacaspase 2. *FEBS J* **280**, 2608–2621 (2013).
7. Gilio, J. M. *et al.* Processing of metacaspase 2 from Trypanosoma brucei (TbMCA2) broadens its substrate specificity. *Biochim Biophys Acta Proteins Proteom* **1865**, 388–394 (2017).
8. Watanabe, N. & Lam, E. Calcium-dependent activation and autolysis of Arabidopsis metacaspase 2d. *J. Biol. Chem.* **286**, 10027–10040 (2011).
9. Watanabe, N. & Lam, E. Two Arabidopsis metacaspases AtMCP1b and AtMCP2b are arginine/lysine-specific cysteine proteases and activate apoptosis-like cell death in yeast. *J Biol Chem* **280**, 14691–14699 (2005).
10. McLuskey, K. *et al.* Crystal structure of a Trypanosoma brucei metacaspase. *PNAS* **109**, 7469–7474 (2012).
11. Moss, C. X., Westrop, G. D., Juliano, L., Coombs, G. H. & Mottram, J. C. Metacaspase 2 of Trypanosoma brucei is a calcium-dependent cysteine peptidase active without processing. *FEBS Letters* **581**, 5635–5639 (2007).
12. Coll, N. S. *et al.* Arabidopsis Type I Metacaspases Control Cell Death. *Science* **330**, 1393–1397 (2010).

Reviewers' comments:

Reviewer #1 (Remarks to the Author):

I appreciate that the authors have sufficiently addressed the previous concerns. I find the manuscript now suitable for publication

Reviewer #3 (Remarks to the Author):

The authors have addressed most of my concerns satisfactory. Two are however still remaining.

"In addition this manuscript lacks perspective of biological importance of the work and a related conclusion."

"What is the biological relevance of an 8 days long activation process of an enzyme? Such long periods are usually associated with transformation of the structure such as fibril formation and not with conformational change of a single loop? Where are the controls of stability of components after 8 days of measurement. A similar publication on MCA-IV revealed Ca dependent activation mechanism which lasted minutes (Zhu et al., 2020, Nat. Comm.) Hence, a much better explanation of activation measurements and activation mechanism than provided is required to satisfy my concerns."

Please note that both remarks are related. I suggest the authors to at least make a comment on this long maturation/activation process. What is the half-life of this yeast species? Have they missed another player?

REVIEWER

“In addition this manuscript lacks perspective of biological importance of the work and a related conclusion.”

“What is the biological relevance of an 8 days long activation process of an enzyme? Such long periods are usually associated with transformation of the structure such as fibril formation and not with conformational change of a single loop? Where are the controls of stability of components after 8 days of measurement. A similar publication on MCA-IV revealed Ca dependent activation mechanism which lasted minutes (Zhu et al., 2020, Nat. Comm.) Hence, a much better explanation of activation measurements and activation mechanism than provided is required to satisfy my concerns.”

Please note that both remarks are related. I suggest the authors to at least make a comment on this long maturation/activation process. What is the half-life of this yeast species? Have they missed another player?

As mentioned in our previous answer, we agree that it seems unlikely that a biological mechanism (i.e the autoprocesing of CgMCA-I) that is part of a cell death phenomenon could take place in eight days.

Interestingly, in a recent publication by Aryamloo et al, the authors explain that *C. glabrata*, exposed to different concentrations of caspofungin, exhibited the features of late apoptosis/necrosis after 18 h of incubation. Whereas this does not mean that metacaspase is necessarily involved in this phenomenon, it does indicate at a minimum that the apoptotic phenomenon in *C. glabrata* is relatively short and inferior to one day.

We hypothesize that the time-scale of this process could possibly be explained by the fact that these experiments were performed *in vitro* and not *in vivo*. It cannot be excluded that in living cell, other protein partners, metacaspase or Ca²⁺ local concentrations, specific cellular localization ... could accelerate the process.

Regarding the “stability of the components after 8 days of measurement”; we did not perform additional experiments to verify the stability of the metacaspase on a longer time scale. However, it should be noticed that when we performed thermostability- and calcium affinity experiments, we observed the exact same affinity for calcium 24 hours after day 0 and 24 hours after day 8, meaning that even after nine days, the enzyme does not seem to degrade.

More strikingly, as mentioned in the Material and Method for the crystallization assay, the droplets were made with enzymes at maturation day 0 in the presence of calcium for CgMCA-ICa, and at maturation day 8 in the presence of magnesium for CgMCA-IMg. What we did not mention in the original manuscript is that it takes several weeks for the crystals to appear and growth (now added to

paragraph “Crystallization, data collection and structure determination” in Material and Method (lines 505-506): “...with Mg^{2+} (CgMCA- I^{Mg}). **Crystals appeared in both conditions after three months. Crystals were cryo-protected...**”. This indicates that, at least under conditions used for crystallization in this study, the protein is stable over a very long period of time in solution.

Altogether, we agree that the 8 days maturation time could be due to the experimental conditions and that further *in vivo* studies could hopefully provide us information regarding the cellular process at stake.

Therefore, we have added the following sentences at the end of the discussion (lines 412-425).

“Surprisingly, during this study, we observed a very long period of maturation (eight days without calcium and four days in presence of calcium Figure 1B). It seems unlikely that a biological mechanism (i.e the autoprocessing of CgMCA-I) that is part of a cell death phenomenon could take place over such a long period of time. In the study of AtMC-IV (Zhu et al., 2020), the authors found that the plant metacaspase shows complete maturation after 10 minutes in presence of 12.5 mM Ca^{2+} . Interestingly, in a recent publication by Aryamloo et al, it was observed that *C. glabrata*, exposed to different concentrations of caspofungin (an antifungal drug), exhibited the features of late apoptosis/necrosis after 18 h of incubation. Whereas this does not mean that metacaspase is necessarily involved in this phenomenon, it does indicate at a minimum that the apoptotic phenomenon in *C. glabrata* is relatively short and inferior to one day. Hence, the length of this process in our study could be due to these experiments being performed *in vitro* and not *in vivo*. It cannot be excluded that in the living cell, other protein partners, metacaspase or Ca^{2+} local concentrations or specific cellular localization could accelerate the process.”

REVIEWERS' COMMENTS:

Reviewer #3 (Remarks to the Author):

The authors' response seems OK to me.